# Predicting hypertension and identifying most important factors among married women in Bangladesh using machine learning approach

Novel Chandra Das[1]*, Probir Kumar Ghosh[1], Md. Alamgir Hossain[1], Uddip Acharjee Shuvo[2], Nipa Rani Talukder[3], Fatema Khatun[1], Mohammad Ziaul Islam Chowdhury[4,5,6]

1 International Centre for Diarrhoeal Disease Research, Dhaka, Bangladesh, 2 Institute of information technology, University of Dhaka, Dhaka, Bangladesh, 3 Department of Computer Science and Engineering, North East University, Dhaka, Bangladesh, 4 Department of Psychiatry, University of Calgary, Hospital Drive NW, Calgary, Canada, 5 Provincial Research Data Services, Alberta Health Services, Alberta, Canada, 6 Department of General Educational Development, Daffodil International University, Dhaka, Bangladesh

* Novel.das@icddrb.org

## Abstract

### Introduction

Hypertension is a leading contributor to maternal and cardiometabolic morbidity in Bangladesh. We developed and interpreted machine-learning (ML) models to predict hypertension and rank associated factors among married women with the goal of informing targeted screening and policy in low-resource settings.

### Methods

We analyzed 4,253 married women from the nationally representative BDHS 2017–18 survey (hypertension prevalence: 23.1%). Twelve ML algorithms were trained under six class-balancing strategies with hyperparameters tuned via random search. Validation used a hold-out test set (80/20) and repeated stratified k-fold cross-validation; bootstrap confidence intervals were estimated for the selected model. Model performance was compared with parametric and non-parametric tests. To interpret results, SHAP was used to rank the top 20 predictors and visualize feature effects. Models quantify associations rather than causation.

### Results

The Extra Trees classifier with SMOTE+ENN achieved the best discrimination (F1 = 0.94; AUC-PR = 0.95; ROC-AUC = 0.95). Compared with the original imbalanced training, minority-class detection improved substantially (Extra Trees F1 increased from 0.08 to 0.94; recall from 0.04 to 0.95) while accuracy and ROC-AUC remained relatively stable across samplers. Statistical testing favored SMOTE+ENN for recall,

**Data availability statement:** The data underlying this study are from the 2017–18 Bangladesh Demographic and Health Survey (BDHS). These data are owned by the DHS Program and cannot be shared by the authors due to third-party restrictions. However, they are publicly available free of charge to all researchers upon registration and approval through the DHS Program website. Access is typically granted within 1–2 business days. The dataset can be requested at: Visit the DHS Program data portal: https://dhsprogram.com/data/ Navigate to the dataset page: https://dhsprogram.com/data/dataset/Bangladesh_Standard-DHS_2017.cfm The analytical codes used in this study, developed in Python and R, are publicly available on GitHub at: https://github.com/NobleNovel/Hypertension_Maternal.

**Funding:** The author(s) received no specific funding for this work.

**Competing interests:** The authors have declared that no competing interests exist.

F1, G-mean and AUC-PR. SHAP identified age, parity, recent births, contraceptive use, spousal education and BMI as key predictors. Younger age (<35 years) and normal/underweight status were protective, while parity ≥2–3, husbands' age ≥40 years and overweight/obesity increased risk.

## Conclusions

An interpretable ensemble model built primarily on sociodemographic and behavioral variables supplemented by limited biometric markers (BMI, glucose) can accurately flag hypertensive risk among married women in Bangladesh. Findings support programmatic integration of risk scores into eRegistries, routine blood pressure checks in family planning and postpartum visits, husband-focused education/SMS interventions and prioritization of high-parity households in high-risk regions. External validation on BDHS-2022 is planned to assess generalizability.

## Introducvtion

Hypertension is a major contributor to cardiovascular disease and chronic kidney disease, two of the leading causes of death and disability worldwide [1–4]. Globally, an estimated 1.13 billion people are affected with 66.7% residing in low- and middle-income countries (LMICs) where prevalence is rising at an alarming pace, particularly in Asia and Southeast Asia [1,3]. In Bangladesh, prevalence among adults aged ≥35 years has nearly doubled, increasing from 25.7% to 48% in recent decades [5]. Gender differences are striking: women, especially those ≥35 years, show a prevalence of 45% compared to 34% in men, representing a marked rise since 2011 [6–12]. Several nationwide studies confirm that women are disproportionately affected with one reporting 28.9% of women hypertensive versus 23.5% of men [13]. Chronic hypertension in women carries serious implications including higher risks of maternal and neonatal complications [14]. Importantly, multiple studies highlight that married women face even higher hypertension rates than unmarried or never-married women, reflecting the interplay of reproductive demands, economic responsibilities, psychosocial stressors and healthcare disparities [15–18].

Machine learning (ML), a key component of artificial intelligence (AI) has emerged as a transformative tool in healthcare. Unlike traditional regression models, which rely on predefined assumptions of linearity and limited variable interactions, ML methods can process high-dimensional data, capture nonlinear relationships and rank the relative importance of predictors. These advantages have allowed ML to consistently outperform conventional approaches in disease prediction, particularly when datasets are complex or involve interrelated risk factors. At the same time, ML faces limitations: many models require large sample sizes for stability are sensitive to data imbalance and if not properly explained, may be viewed as "black boxes." Addressing interpretability and ensuring fairness remain essential for public health adoption [19–23].

In cardiovascular research, ML has been applied successfully to echocardiogram analysis and risk prediction for acute decompensated heart failure with models such as K-Nearest Neighbor (KNN), Support Vector Machines (SVM) and ensemble methods achieving strong predictive accuracy [24,25].

For hypertension specifically, ML has demonstrated clear advantages. Tree-based algorithms like random forest (RF) and extreme gradient boosting (XGBoost) have outperformed regression-based methods with XGBoost achieving AUROC values ranging from 0.766 to 1.00 across datasets including 0.894 in semi-laboratory settings when ranking predictors such as systolic blood pressure, waist circumference and albumin levels [26–33]. Recursive feature elimination (RFE) further enhances these models by systematically refining the set of predictors. Comparative studies also confirm that ML surpasses Cox and logistic regression in larger and more complex datasets [34,35]. Hybrid approaches, such as combining RFE with XGBoost, have achieved superior accuracy while models applied to electronic health records capture dynamic features often missed by traditional statistics [36–39].

The utility of ML extends beyond hypertension. In oncology, artificial neural networks and Bayesian networks stratify patients into risk categories while in obesity research, classifiers such as SVMs and quadratic discriminant analysis outperform logistic regression by detecting nonlinear behavioral patterns [40,41]. Nutritional epidemiology studies show k-nearest neighbors and random forests classify cardiometabolic risk more effectively than linear regression [42]. Ensemble models are particularly advantageous in smaller datasets, such as South African studies predicting abnormal angiograms where they outperformed traditional statistical approaches [43]. ML has also been employed for imputing missing data, which improved breast cancer recurrence prediction and for enhancing cerebral ischemia outcome prediction in aneurysmal subarachnoid hemorrhage patients [44,45]. For coronary heart disease survival, SVMs achieved high accuracy [46] while neural networks such as multilayer perceptrons (MLP) and radial basis function (RBF) networks outperformed other classifiers in predicting essential hypertension, demonstrating their ability to capture complex and nonlinear relationships [47]. Notably, gradient boosting methods with RFE outperformed Cox regression and recalibrated Framingham Risk Scores in predicting adverse outcomes in young hypertensive patients, achieving a C-statistic of 0.757 [48].

One persistent challenge in ML health research is data imbalance, which can reduce sensitivity and lead to misclassification of minority outcomes. Methods such as Synthetic Minority Oversampling Technique (SMOTE), random undersampling (RUS) and cost-sensitive learning have been widely applied to mitigate this issue [36,49–56]. For example, a cost-sensitive deep neural network improved mortality prediction in acute myocardial infarction patients with hypertension by 2.58% AUC compared to ensemble models [57]. SVM models with SMOTE increased accuracy from 91% to 98% [58] while a RUS-applied random forest improved stroke risk prediction among hypertensive adults, yielding AUC 0.624 and sensitivity 63.9% [59].

Interpretability is equally critical. SHAP (Shapley Additive Explanations) has emerged as a powerful, model-agnostic framework that provides consistent and transparent feature attribution [60–65]. Unlike LASSO or ANOVA, SHAP captures nonlinearities and interaction effects, offering granular, instance-level explanations. Visualization tools enhance communication with clinicians, and SHAP's flexibility in handling missing and imbalanced data makes it highly suitable for real-world datasets [62,65–69]. Moreover, its support for multi-omics integration and human–machine collaboration further enhances its utility in personalized healthcare.

Despite these advances, significant gaps remain. Most hypertension-focused ML studies in South Asia including Bangladesh have concentrated on general adult populations with little attention to married women, who face distinct risks shaped by reproductive roles, domestic workloads and socio-economic constraints. Subgroup-specific validation remains limited, and few studies explicitly integrate cultural and gender-related determinants into predictive models. Although advanced methods such as SMOTE enhance calibration and sensitivity [70–76], they have rarely been applied to this subgroup in Bangladesh. Furthermore, BDHS-2022 was released after our analysis, that is why we trained the model on BDHS-2017–18 and pre-specified external validation on BDHS-2022. This approach ensures robust assessment of generalizability without altering the model development process based on a single additional dataset. Against this backdrop,

the present study aims to develop ML models to identify predictive factors of hypertension among married women in Bangladesh. To our knowledge, this is the first study in Bangladesh to apply an extensive set of algorithms combined with class-balancing techniques to this population, contributing both methodological innovation and population-specific insights. Therefore, the objective of this study is to develop and validate interpretable machine-learning models for predicting hypertension among married women in Bangladesh, integrating sociodemographic, behavioral and biometric factors (such as BMI and diabetes status) to identify the most influential predictors and provide evidence to guide targeted screening and public health interventions in low-resource settings

## Methodology

### Data sources

We have used Bangladesh Demographic and Health Survey (BDHS) 2017−18 data in this study, which is the nationally representative survey. The National Institute of Population Research and Training, Medical Education and Family Welfare Division and Ministry of Health and Family Welfare jointly conducted the survey from October 2017 to March 2018.

### Sampling method and sample size/study population and survey design

The Bangladesh Bureau of Statistics of the 2011 Population and Housing Census of the People's Republic of Bangladesh provided a complete list of enumeration areas (EAs) covering the whole residing population in Bangladesh, which was used in a survey to determine the sampling frame for the 2017−18 BDHS. The survey employed a two-stage stratified cluster sampling as a sampling method where, in the 1st stage of sampling, 675 enumeration areas (EAs) were chosen, whereas 250 EAs were from urban areas and 425 from rural areas with a probability proportional to the EA scale and then a systematic sample of 30 households per EA was chosen in the 2nd stage of sampling to provide statistically accurate estimates of key demographic and health variables for the nation as a whole, rural and urban areas separately and each of the eight divisions. Finally, after selecting 20,250 residential households, approximately 20,100 ever-married women aged 15–49 were expected to complete the interviews [27]. At last, 19,457 households were successfully interviewed and 5,138 women underwent blood pressure and blood glucose measurements. From the 4,546 married women, 4,253 women were considered for final analysis after the termination of pregnant women and deleting missing value or missing information (Fig 1).

In this analytic sample of 4,253 married women, the prevalence of hypertension was about 23.1% (n ≈ 982) ensuring an adequate number of positive events for training supervised ML models. This sample size provides sufficient power for developing and evaluating predictive algorithms in imbalanced health data contexts, consistent with prior methodological recommendations [77–79].

### Dependent feature

Hypertension was considered if the participant's systolic blood pressure was ≥ 140 mmHg or the diastolic blood pressure was ≥ 90 mmHg or if the person had been taking prescribed medicine to lower blood pressure [80].

### Independent feature

The independent variables in the study are considered from the previous related literature [81–83]. In this study, we considered administrative division of Bangladesh (Barisal, Chittagong, Dhaka, Khulna, Mymensingh, Rajshahi, Rangpur,Sylhet) type of place of residence (urban, rural) respondents highest educational level (no education, primary, secondary and higher)husband/partner educational level (no education, primary, secondary and higher), unmet need for contraception (unmet need for spacing, unmet need for limiting, using for spacing, using for limiting, no unmet need, infecund,

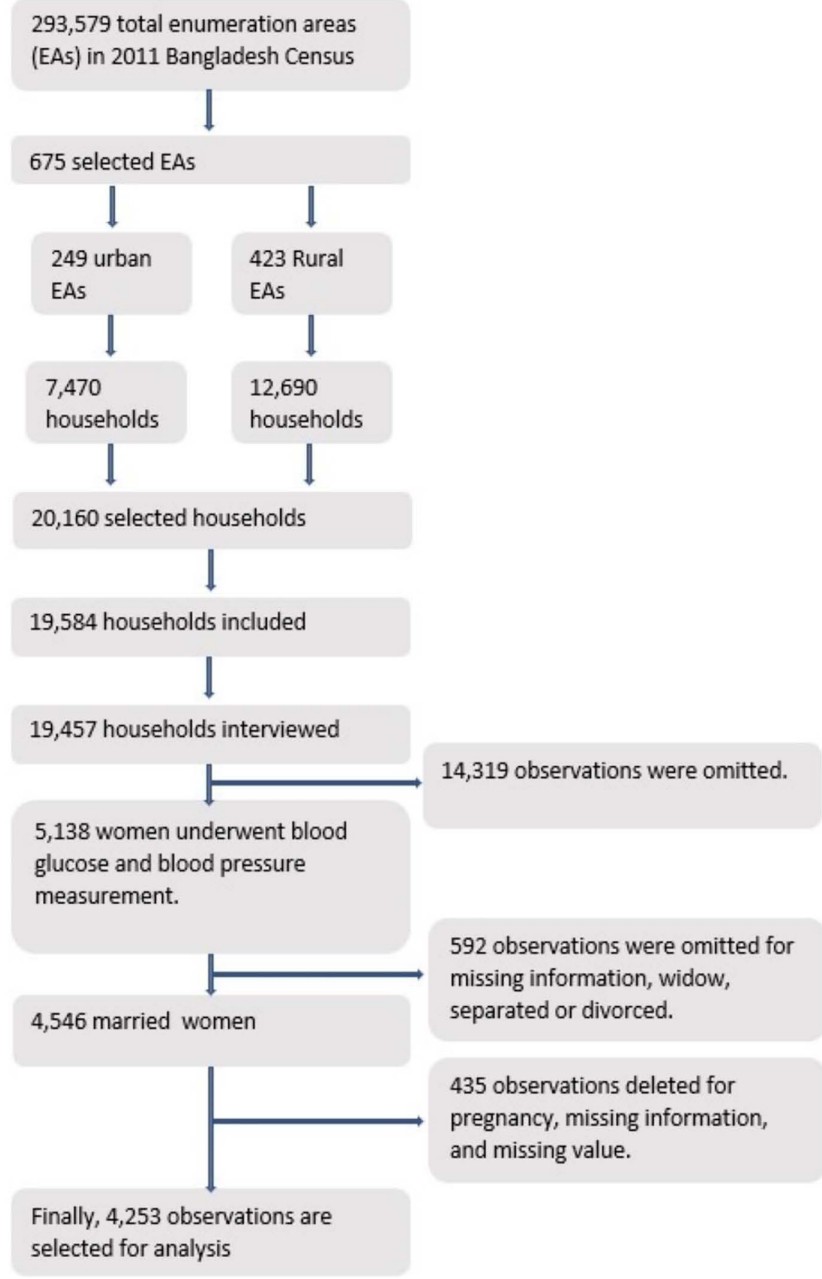

**Fig.1. Data selection flow chart.**

menopausal) religion (Islam, Hinduism, Buddhism, Christianity) sex of household head (male, female) wealth index combined (poorest, poorer, middle, richer, richest), current use by method type (no method, folkloric method, traditional method, modern method) currently amenorrhoeic (yes, no) currently abstaining (yes, no) currently residing with husband/ partner (living with her, staying elsewhere) household members (< 4 persons and ≥4 persons) respondent's occupation (working, not working) number of living children (no living children, one, two and more than two children) husband/

partner's occupation (working, not working) respondent's current age (less than 35 years, 35–40 years and above 40 years old) husband/partner's age (less than 35 years, 35–40 years and above 40 years old) total children ever born (no children ever born, one, one to three and above three children born) births in last five years (no birth, one and above one) daughters who have died (no died, at least one died) sons who have died (no died, at least one died) age difference between husband/partner and wife (less than ten year, ten and above).

### Derived variables

**Diabetes status.** Fasting plasma glucose (FPG) was considered to calculate diabetes. The HemoCue Glucose 201 DM system with plasma conversion was used to test a drop of capillary blood obtained from consenting eligible respondents from the middle or ring finger. The system automatically converted the fasting whole blood glucose measurements taken in the survey to FPG equivalent values [84,85]. To classify diabetes World Health Organization (WHO) criteria were used [86]. Diabetes was considered if the FPG level was greater than or equal to 7 mmol/l or self-reported diabetes medication use.

**Body Mass Index (BMI).** Calculated as weight in kilograms divided by height in meters squared (kg/m²). Categories were defined according to WHO cut-offs: underweight (<18.5), normal (18.5–24.9), overweight (25.0–29.9) and obese (≥ 30) [87].

### Feature selection

All sociodemographic, behavioral, biometric and anthropometric variables with theoretical or empirical relevance to hypertension were extracted from the BDHS 2017–18 dataset. No additional feature engineering or automated feature selection algorithms (e.g., Boruta, LASSO, or recursive feature elimination) were applied. Instead, variable inclusion was guided by existing epidemiological literature and prior BDHS-based hypertension studies.

### Data preparation

**Missing data.** Cases with missing values were excluded. No imputation was performed to avoid introducing artificial variability.

### Survey weights

The BDHS 2017–18 employs a complex survey design with stratification, clustering and sampling weights to ensure national representativeness. In the present study, we did not apply survey/sample weights because the primary aim was methodological focused on evaluating and comparing the predictive performance of class balancing approaches integrate with machine learning algorithms and ranked risk and protected features rather than estimating population-level prevalence or nationally representative parameters. This approach is consistent with prior ML studies using DHS data in similar contexts [73,88].

### Feature scaling

Data normalization; which is a process of re-scaling the feature value, is very important because most of the machine learning algorithms use Euclidean distance between two data points as a distance metrics, so without Feature scaling, the machine learning algorithms may not execute properly [89]. For rescaling, standardization technique has been applied which the mean is zero and the standard deviation is one.

### Encoding

Categorical variables were transformed using one-hot encoding to allow use in ML algorithms.

 

## Imbalanced data problem

As imbalanced data lead to the majority class dominates minority class that's why the, it impacts the reliability of determinations from the dataset, algorithm biased towards majority class and may provide more inaccurate result [90–92]. It is found that for balanced data (where classes proportion are equal) may lead extract best result for identifying the factors. To convey the issue of imbalanced data, Synthetic Minority Oversampling Techniques (SMOTE), Adaptive Synthetic Sampling (ADASYN), Tomek Links (TLs), Edited Nearest Neighbor (ENN), SMOTE-TomekLinks, SMOTE-ENN techniques are applied to resolve the issue (see supplementary File S1 Appendix).

## Machine learning algorithms

We evaluated 12 algorithms: Logistic Regression, Decision Tree, K-Nearest Neighbors (KNN), Random Forest, Extra Trees, AdaBoost, Gradient Boosting Machine (GBM), XGBoost, LightGBM, CatBoost, Support Vector Machine (SVM) and Multilayer Perceptron (MLP).

We included Extra Trees as it is computationally efficient, less prone to overfitting in small subgroups and yields stable feature importance rankings, complementing RF and XGBoost [93–95]. Detailed algorithm descriptions are in Supplementary S2 Appendix.

## Model training with parameter optimization

**Hold-out cross validation.** Original dataset is divided into training and testing subsets where 80% data belongs to training and the rest 20% belongs to testing subset.

## Repeated stratified k fold cross validation

The dataset is divided into k-folds, where one of the k-folds is selected as a validation set and the remaining sets comprise the training set. Until each one of them forms validation sets, the operation is repeated for each fold, which means for the n number of repetitions the process will be repeated k × n times.

We avoided overfitting and underfitting problems by employing hold-out cross-validation to split training and testing set and to reduce the sampling error, repeated stratified k-fold cross-validation as a validation method was applied. Additionally, we utilized random search to select model parameters using hyper-parameters as there is more chance to select the best parameter [96].

We note that external validation was not possible as BDHS 2022 data was not fully accessible at the time of analysis. This remains a priority for future work to assess model generalizability.

## Evaluation methods

To assess the performance of machine learning methods, we employed confusion matrix, Matthews correlation coefficient, Cohens-kappa, F1-score, G-mean, recall/ sensitivity, specificity, accuracy, precision, AUC-ROC, AUC-PR. For the evaluation of the matric score we used the Anderson-Darling test to check data normality, One-way repeated measure ANOVA was utilized to determine the overall difference among class-balancing techniques, Tukey's HSD test was used to classify the significance difference among specific group of class balancing techniques, Friedman test was used to find the difference among group instead of one-way ANOVA where normality assumptions are violated (supplementary File S3 Appendix).

## External validation plan

At the time of model development, BDHS-2022 data was not publicly released; consequently, model training and internal validation used BDHS-2017/18. BDHS-2022 is now available and differs modestly in feature scope; to avoid design leakage,

we will treat BDHS-2022 strictly as an external test set. Given the scarcity of comparable datasets of the same type, we also reserve BDHS-2022 for validating the deployed system. No retraining or feature re-specification will be performed for the external test; instead, we will apply identical preprocessing and the pre-specified decision threshold. This analysis will provide an out-of-sample assessment of generalizability across a later, post-COVID cohort and any instrument changes.

### Software and hardware

Data preprocessing was performed in STATA 15, ML models in Python (scikit-learn, XGBoost, CatBoost, LightGBM, TensorFlow etc.) and statistical tests in R. Training was carried out on a workstation with Intel Core i5-8365U CPU (1.60 GHz) and 8 GB RAM. While resource limitations constrained deeper architectures and exhaustive grid searches, the chosen algorithms were successfully optimized within these constraints.

### Multicollinearity

To assess multicollinearity, we examined variance inflation factors (VIF) and pairwise correlations among predictors. As expected with one-hot encoded categorical variables, VIF values were inflated to infinity due to perfect linear dependence between dummy categories. Because our primary models were gradient-boosted decision trees, which are robust to correlated features, this did not compromise predictive validity. For clarity, we also inspected pairwise Spearman correlations among key predictors, focusing on socio-demographic and fertility-related variables, which often show natural dependencies.

### Ethical considerations

This study utilized publicly available secondary data from the Bangladesh Demographic and Health Survey (BDHS), which is conducted by the ICF and the Bangladesh Medical Research Council (BMRC). Prior to the data collection, ethical approval was obtained from the Institutional Review Board (IRB) of ICF, USA, and the National Research Ethics Committee of the BMRC. Informed written consent was obtained from all participants involved in the original survey.

As this study involves secondary data analysis, we obtained permission to access the de-identified data from the DHS Program. Since the data was de-identified and publicly available, no additional ethical approval was required for this analysis. The study adhered to the relevant guidelines and regulations for secondary data use.

We additionally highlight that fairness and bias are critical in ML health research. Sensitive variables were treated with caution and SHAP interpretability was used to ensure transparency in feature attribution. These safeguards support responsible use of advanced ML in public health policy.

### Study workflow

An overview of the analytical workflow from data extraction and preprocessing through class balancing, model training, hyperparameter tuning, validation and evaluation, followed by statistical analysis to compare class-balancing techniques and model performance and SHAP-based interpretation is presented in Fig 2. This schematic provides a concise visual summary of the methodological pipeline described in the preceding subsections.

## Result

### Outcome characteristics

Among 4,253 married women, 23.1% had hypertension and 76.9% were normotensive.

### Socio-demographic and clinical characteristics of the study participants

Participants were predominantly rural (64.7%). Wealth distribution was approximately even across quintiles (poorest 19.5% to richest 21.9%). Educational attainment was most commonly secondary for women (37.7%) and primary for

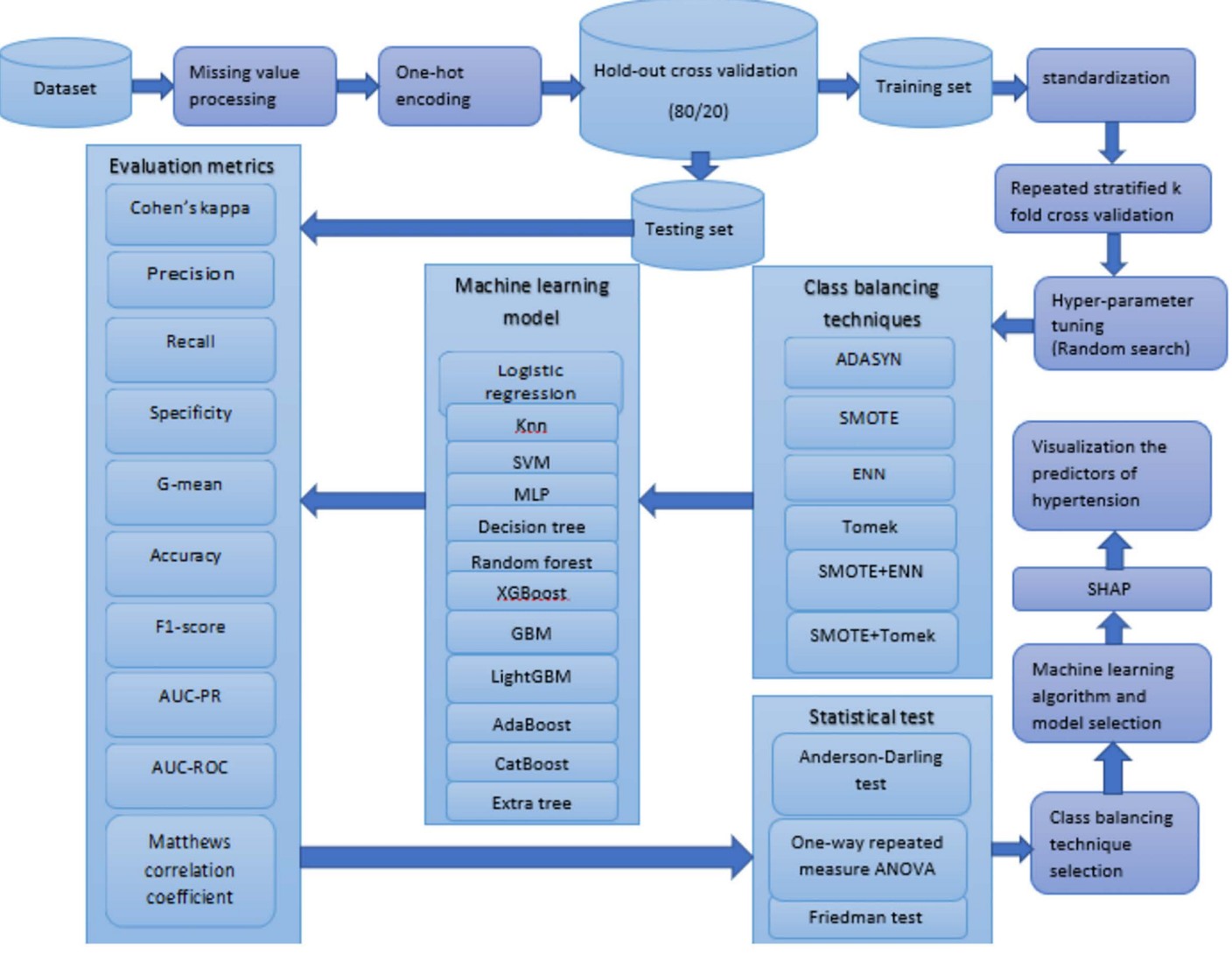

**Fig 2. Workflow of machine learning pipeline for hypertension prediction and risk factor ranking of married women in Bangladesh.**

husbands/partners (32.7%). Median respondent age was 31 years (IQR 25–39); median husband/partner age was 40 years (IQR 32–48). Most respondents were <35 years (61.1%) whereas 41.9% of husbands/partners were >40 years. Fertility profiles showed 51.1% with two to three ever-born children; 59.4% had no births in the past five years. Nutritional status was: underweight 11.7%, normal 53.9%, overweight 27.4%, and obese 7.0. Glycemic categories were: normoglycemia 77.8%, intermediate hyperglycemia 13.1%, hyperglycemia 7.8% and hypoglycemia 1.3% (Table 1).

### Stratification of hypertension among married women in Bangladesh

Table 1 demonstrates that Prevalence varied by division, highest in Rangpur (27.8%) and lowest in Mymensingh (18.8%). Urban residence was associated with higher prevalence than rural (24.1% vs. 22.5%). Prevalence decreased with higher female education (no education 31.3%, higher education 18.9%) and was elevated when husbands had no education (26.6%). Hypertension rose across wealth quintiles (poorest 18.7% to richest 27.6%). Reproductive status

**Table 1. Demographic and clinical characteristics of study participants.**

| Characteristics | Total | Hypertension | 95% CI |
|---|---|---|---|
| **Administrative division of Bangladesh** | | | |
| Barisal | 453 (10.7) | 115 (25.4) | 21.6–29.6 |
| Chittagong | 588 (13.8) | 160 (27.2) | 23.8–30.9 |
| Dhaka | 579 (13.6) | 114 (19.7) | 16.7–23.1 |
| Khulna | 588 (13.8) | 136 (23.1) | 19.9–26.7 |
| Mymensingh | 467 (10.9) | 88 (18.8) | 15.6–22.6 |
| Rajshahi | 580 (13.6) | 125 (21.6) | 18.5–25.0 |
| Rangpur | 551 (12.9) | 153 (27.8) | 24.2–31.7 |
| Sylhet | 447 (10.5) | 90 (20.1) | 16.6–24.1 |
| **Type of place of residence** | | | |
| Urban | 1,503 (35.3) | 362 (24.1) | 21.9–26.5 |
| Rural | 2,750 (64.7) | 619 (22.5) | 20.8–24.3 |
| **Respondent's highest educational level** | | | |
| No education | 689 (16.2) | 216 (31.3) | 27.9–34.9 |
| Primary | 1,372 (32.3) | 332 (24.2) | 21.9–26.7 |
| Secondary | 1,605 (37.7) | 322 (20.1) | 18.2–22.2 |
| Higher | 587 (13.8) | 111 (18.9) | 15.9–22.3 |
| **Husband/partner educational level** | | | |
| No education | 943 (22.2) | 251 (26.6) | 23.8–29.5 |
| Primary | 1,391 (32.7) | 288 (20.7) | 18.6–22.9 |
| Secondary | 1,240 (29.2) | 291 (23.5) | 21.2–26.0 |
| Higher | 679 (15.9) | 151 (22.2) | 19.2–25.5 |
| **Unmet need for contraception** | | | |
| Unmet need for spacing | 172 (4.0) | 15 (8.7) | 5.3–14.0 |
| Unmet need for limiting | 257 (6.0) | 59 (23.0) | 18.1–28.8 |
| Using for spacing | 723 (17.0) | 69 (9.9) | 7.9–12.4 |
| Using for limiting | 2,124 (49.9) | 574 (27.1) | 25.2–29.1 |
| No unmet need | 435 (10.2) | 63 (14.5) | 11.5–18.0 |
| Infecund/menopausal | 542 (12.7) | 201 (37.1) | 33.2–41.2 |
| **Religion** | | | |
| Islam | 3,813 (89.7) | 855 (22.4) | 21.1–23.7 |
| Hinduism | 403 (9.5) | 114 (28.3) | 24.2–32.8 |
| Buddhism | 26 (0.6) | 8 (30.8) | 16.2–51.6 |
| Christianity | 11 (0.3) | 4 (36.4) | 14.5–66.6 |
| **Sex of household head** | | | |
| Male | 3,724 (87.6) | 859 (23.1) | 21.8–24.4 |
| Female | 529 (12.4) | 122 (23.1) | 19.8–26.8 |
| **Wealth index combined** | | | |
| Poorest | 827 (19.5) | 154 (18.7) | 16.2–21.6 |
| Poorer | 833 (19.6) | 176 (21.1) | 18.4–24.0 |
| Middle | 837 (19.7) | 194 (23.2) | 20.5–26.1 |
| Richer | 821 (19.3) | 199 (24.2) | 21.4–27.3 |
| Richest | 935 (21.9) | 258 (27.6) | 24.7–30.7 |
| **Current use of contraceptive methods** | | | |
| No method | 1,406 (33.1) | 338 (24.1) | 21.9–26.4 |
| Folkloric method | 14 (0.3) | 4 (28.6) | 11.7–55.3 |

*(Continued)*

| Characteristics | Total | Hypertension | 95% CI |
|---|---|---|---|
| Traditional method | 455 (10.7) | 126 (27.7) | 23.7–32.1 |
| Modern method | 2,378 (55.9) | 513 (21.6) | 20.0–23.4 |
| **Currently amenorrhoeic** | | | |
| Yes | 4,058 (95.4) | 948 (23.4) | 22.1–24.8 |
| No | 195 (4.6) | 33 (18.5) | 13.5–24.8 |
| **Currently abstaining** | | | |
| Yes | 4,112 (96.7) | 955 (22.2) | 20.9–23.6 |
| No | 141 (3.3) | 26 (18.5) | 12.9–26.0 |
| **Currently residing with husband/partner** | | | |
| Living with her | 3,564 (83.8) | 848 (23.8) | 22.4–25.2 |
| Staying elsewhere | 689 (16.2) | 133 (19.3) | 16.5–22.5 |
| **Household members** | | | |
| <4 persons | 849 (19.9) | 221 (26.1) | 23.1–29.4 |
| ≥4 persons | 3,404 (80.1) | 760 (22.3) | 21.0–23.8 |
| **Employment status of respondent** | | | |
| Working | 2,231 (52.5) | 528 (23.7) | 21.9–25.6 |
| Not working | 2,022 (47.5) | 453 (22.4) | 20.6–24.3 |
| **Number of living children** | | | |
| No living children | 275 (6.5) | 35 (12.7) | 9.3–17.1 |
| One | 926 (21.8) | 118 (12.7) | 10.7–15.1 |
| Two | 1,405 (33.0) | 316 (22.5) | 20.3–24.9 |
| More than two | 1,647 (38.7) | 512 (31.1) | 28.8–33.6 |
| **Employment status of husband/partner** | | | |
| Working | 4,173 (98.1) | 949 (22.7) | 21.4–24.0 |
| Not working | 80 (1.9) | 32 (40.0) | 29.9–51.2 |
| **Respondent's current age** | | | |
| <35 years | 2,598 (61.1) | 352 (13.5) | 12.2–15.0 |
| 35–40 years | 783 (18.4) | 265 (33.9) | 30.6–37.3 |
| >40 years | 872 (20.5) | 364 (41.7) | 38.4–45.1 |
| **Husband/partner's age** | | | |
| <35 years | 1,229 (28.9) | 113 (9.2) | 7.7–11.0 |
| 35–40 years | 1,240 (29.2) | 243 (19.6) | 17.4–22.0 |
| >40 years | 1,784 (41.9) | 625 (35.1) | 32.9–37.5 |
| **Total children ever born** | | | |
| No children ever born | 268 (6.3) | 31 (11.6) | 8.3–16.0 |
| One | 844 (19.8) | 100 (11.8) | 9.7–14.0 |
| Two to three | 2,174 (51.1) | 548 (25.2) | 23.5–27.1 |
| Above three | 967 (22.7) | 302 (31.2) | 28.3–34.3 |
| **Births in last five years** | | | |
| No birth | 2,526 (59.4) | 755 (29.9) | 27.9–32.0 |
| One | 1,456 (34.2) | 193 (13.3) | 11.6–15.3 |
| Above one | 271 (6.4) | 33 (12.2) | 8.8–16.7 |
| **Daughters who have died** | | | |
| None | 3,918 (92.1) | 887 (22.6) | 21.3–23.9 |
| ≥1 daughter | 335 (7.8) | 94 (28.1) | 23.6–33.2 |

*(Continued)*

**Table 1.** (Continued)

| Characteristics | Total | Hypertension | 95% CI |
|---|---|---|---|
| **Sons who have died** | | | |
| None | 3,827 (89.9) | 859 (22.4) | 21.1–23.7 |
| ≥1 son | 426 (10.1) | 122 (28.6) | 24.5–33.2 |
| **Age difference between husband/partner and wife** | | | |
| <10 years | 3,001 (70.6) | 667 (22.2) | 20.8–23.6 |
| ≥10 years | 1,252 (29.4) | 314 (25.1) | 22.7–27.7 |
| **Body mass index (BMI)** | | | |
| Underweight | 497 (11.7) | 55 (11.1) | 8.5–14.3 |
| Normal | 2,295 (53.9) | 406 (17.7) | 16.2–19.4 |
| Overweight | 1,163 (27.4) | 395 (34.0) | 31.3–36.9 |
| Obese | 298 (7.0) | 125 (41.9) | 36.5–47.5 |
| **Diabetes** | | | |
| Hypoglycemia | 56 (1.3) | 11 (19.6) | 11.5–31.7 |
| Normoglycemia | 3,307 (77.8) | 706 (21.3) | 19.9–22.7 |
| Intermediate hyperglycemia | 558 (13.1) | 131 (23.5) | 20.3–27.0 |
| Hyperglycemia | 332 (7.8) | 133 (40.1) | 34.9–45.6 |

mattered: women who were infecund/menopausal had 37.1% prevalence; those using contraception to limit births had 27.1% whereas unmet need for spacing had the lowest (8.7%).

Age gradients were pronounced:<35 years 13.5%, 35–40 years 33.9% and >40 years 41.7%. Husband/partner age>40 years was also associated with higher prevalence (35.1%). BMI showed a graded pattern: underweight 11.1%, normal 17.7%, overweight 34.0%, obese 41.9%. By religion, the highest stratum was among Christians (36.4%) and the lowest among Muslims (22.4%), acknowledging small denominators in minority groups. Households with >2 living children had higher prevalence (31.1%). Participants whose husbands/partners were not working had higher prevalence (40.0%) versus those with employed partners (22.7%), noting the small size of the unemployed group (n = 80). Smaller households (<4 persons) showed higher prevalence (26.1%) than larger households (22.3%).

## Class distribution before and after class balancing

To ensure transparency in preprocessing, we report class distributions in the training data before and after class balancing. The original training set contained 3,581 women, with 827 (23.1%) hypertensive and 2,754 (76.9%) normotensive. Both SMOTE and ADASYN achieved perfect balance (50/50). Tomek Links modestly increased the minority proportion to 24.7% by removing borderline majority cases, while ENN shifted the minority share to 35.0%. Hybrid methods produced stronger changes: SMOTE+Tomek restored exact balance (50/50), whereas SMOTE+ENN yielded a minority-dominant distribution (63.3% hypertensive). These comparisons highlight trade-offs between balance and retained sample size across class balancing strategies (Supplementary Table S1 in S4 Appendix).

## Parameter optimization

Model performance shifted with sampling technique. Logistic Regression performed best with L2 (C = 11.29; lbfgs) on the actual data, but under SMOTE/ADASYN/TomekLinks/ENN required lower C, class weighting, or L1 with saga. Extra Trees favored 200 estimators, log2 features, and entropy on actual data; SMOTE reduced trees and used Gini, while ADASYN/TomekLinks retained entropy and increased trees. Decision Trees preferred deeper structures/smaller leaves under SMOTE and SMOTE+ENN; ENN favored Gini. AdaBoost benefited from more estimators and higher learning rates

(up to 1.5), often with SAMME.R under ADASYN/ENN. SVMs shifted between RBF and polynomial kernels depending on sampler. XGBoost, LightGBM, KNN, CatBoost, Random Forest, and GBM all exhibited sampler-specific hyperparameter adjustments (Table 2). Unless otherwise noted, cross-model metrics in Table 3 are fold-averaged over stratified 5-fold CV on the training set. For the selected ExtraTrees + SMOTE+ENN model, test-set uncertainty is reported via 95% bootstrap CIs (Table 6).

## Classification efficacy and confusion matrix

Across classifiers, training on the original imbalanced data produced inflated accuracy and specificity but markedly poor minority sensitivity (low recall and F1), evidencing majority-class bias (Table 3; Fig 3). For example, ExtraTrees (Actual) achieved Accuracy = 0.78, Specificity = 1.00, Recall = 0.04 and F1 = 0.08, consistent with its confusion matrix on the original test split (Table 4). After applying SMOTE+ENN, ExtraTrees improved substantially F1 = 0.94, Recall = 0.95, Cohen's κ = 0.79, MCC = 0.79 and G-mean = 0.89 in agreement with the corresponding SMOTE+ENN confusion matrix (Table 5). These matched matrix–metric pairs ensure internal coherence. Cross-model medians (Friedman ranks) indicated that TomekLinks tended to yield the highest Accuracy and Specificity overall (Supplementary Table S2E in S4 Appendix) whereas on the independent test set ExtraTrees with SMOTE+ENN achieved the highest Accuracy among our models (0.91 vs. 0.78 on the original data), underscoring that balancing improved clinically relevant discrimination.

## Global comparison of class-balancing techniques

ML algorithms exhibited varying performance across balancing techniques, motivating formal statistical testing. Except for AUC-PR, most metrics were non-normal by Anderson–Darling (Supplementary S2A Table in S4 Appendix), so ANOVA was restricted to AUC-PR. Repeated-measures ANOVA on AUC-PR showed significant differences across techniques (Supplementary S2B Table in S4 Appendix), and Tukey's HSD identified SMOTE+ENN and ENN as significantly superior for AUC-PR in multiple pairwise contrasts (Supplementary S2C Table in S4 Appendix). For the remaining non-normal metrics, Friedman tests showed significant omnibus differences (Supplementary S2D Table). Post-hoc rank summaries indicated SMOTE+ENN led G-mean, Recall, and F1 (ENN second) while TomekLinks ranked highest for Accuracy, Specificity, Precision, and AUROC (Supplementary Table S2E in S4 Appendix).

Boxplots of AUCPR, F1, Recall, Precision, Specificity, G-mean, Accuracy and AUC-ROC (Fig 4–11) illustrate these trends: SMOTE+ENN achieved the highest Recall, F1 and AUC-PR, whereas the original data scored highest in Accuracy and Specificity.

## Feature importance

Integrating SHAP with a trained model enabled analysis of global and local predictors. Figure 12a shows the most important features; Fig 12b classifies the top 20 factors by direction of association; Fig 12c presents a SHAP summary plot combining feature importance and effects.

In Fig 12a, respondent age < 35 years was the most influential factor, followed by overweight, having two to three ever-born children, and husband/partner with secondary education. Additional contributors included underweight, normal weight, husband/partner < 35 years, > 2 living children, no births in the last five years and obesity.

Fig 12b indicates negative associations (blue) for age < 35, underweight/normal weight, husband/partner < 35 years, respondent age 35–40, one child in last five years, normoglycemia, residence in Rajshahi and no formal education for husband/partner. Positive associations (red) included overweight, two to three ever-born children, husband/partner with secondary education, > 2 living children, no births in five years, obesity, husband/partner >40, traditional contraceptive use, unmet need for limiting, richest quintile and age difference ≥10 years.

Fig 12c shows these effects at the individual level: blue points (low SHAP values) correspond to features decreasing predicted risk, while red/purple points (high SHAP values) correspond to features increasing predicted risk.

**Table 2.** Optimal parameters of the machine learning algorithm.

| ML model | Sampling | Parameter | ML model | Sampling | Parameter |
|---|---|---|---|---|---|
| Logistic Regression | Actual | Penalty = l2, C = 11.288378916846883, Solver = lbfgs, Class_ weight = None, Max _iter = 2500 | Extra Trees | Actual | n_ estimators = 200, max_ features = log2, max_ depth = None, min_ samples_ leaf = 6, min_ samples_ split = 76, bootstrap = True, criterion = entropy |
| | SMOTE | Penalty = l2, C = 1.623776739188721, Solver = liblinear, Class _weight = None, Max _iter = 1000 | | SMOTE | n_ estimators = 52, max_ features = log2, max_ depth = None, min_ samples_ leaf = 8, min_ samples_ split = 16, bootstrap = False, criterion = gini |
| | ADASYN | Penalty = l2, C = 0.012742749857031334, Solver = liblinear, Class _weight = balanced, Max _iter = 5000 | | ADASYN | n_ estimators = 178, max_ features = None, max_ depth = None, min_ samples_ leaf = 2, min_ samples_ split = 10, bootstrap = True, criterion = entropy |
| | TomekLinks | Penalty = l2, C = 0.0018329807108324356, Solver = lbfgs, Class _weight = balanced, Max _iter = 2500 | | TomekLinks | n_ estimators = 115, max_ features = None, max_ depth = None, min_ samples_ leaf = 2, min_ samples_ split = 30, bootstrap = True, criterion = entropy |
| | ENN | Penalty = l1, C = 0.23357214690901212, Solver = saga, Class _weight = None, Max _iter = 2500 | | ENN | n_ estimators = 157, max_ features = None, max_ depth = None, min_ samples_ leaf = 2, min_ samples_ split = 6, bootstrap = True, criterion = entropy |
| | SMOTE+ TomekLinks | Penalty = l2, C = 11.288378916846883, Solver = liblinear, Class _weight = None, Max _iter = 5000 | | SMOTE+ TomekLinks | n_ estimators = 157, max_ features = None, max_ depth = None, min_ samples_ leaf = 6, min_ samples_ split = 18, bootstrap = False, criterion = entropy |
| | SMOTE+ ENN | Penalty = l1, C = 0.23357214690901212, Solver = liblinear, Class _weight = None, Max _iter = 5000 | | SMOTE+ ENN | n_ estimators = 94, max_ features = None, max_ depth = None, min_ samples_ leaf = 6, min_ samples_ split = 6, bootstrap = True, criterion = entropy |
| Decision tree | Actual | Max _depth = 12, Min _samples _split = 44, Min _samples _leaf = 44, Criterion = entropy | AdaBoost | Actual | Algorithm = SAMME.R, n_ estimators = 73, learning_ rate = 0.5 |
| | SMOTE | Max _depth = 28, Min _samples _split = 52, Min _samples _leaf = 4, Criterion = entropy | | SMOTE | Algorithm = SAMME, n_ estimators = 136, learning_ rate = 1.5 |

*(Continued)*

| ML model | Sampling | Parameter | ML model | Sampling | Parameter |
|---|---|---|---|---|---|
| | ADASYN | Max _depth = 16, Min _samples _split = 48, Min _samples _leaf = 6, Criterion = entropy | | ADASYN | Algorithm = SAMME.R, n_ estimators = 178, learning_ rate = 1.5 |
| | TomekLinks | Max _depth = 4, Min _samples _split = 86, Min _samples _leaf = 40, Criterion = entropy | | TomekLinks | Algorithm = SAMME, n_ estimators = 115, learning_ rate = 1.5 |
| | ENN | Max _depth = 8, Min _samples _split = 88, Min _samples _leaf = 32, Criterion = gini | | ENN | Algorithm = SAMME.R, n_ estimators = 136, learning_ rate = 0.5 |
| | SMOTE+ TomekLinks | Max _depth = 22, Min _samples _split = 52, Min _samples _leaf = 4, Criterion = entropy | | SMOTE+ TomekLinks | Algorithm = SAMME.R, n_ estimators = 178, learning_ rate = 1.5 |
| | SMOTE+ ENN | Max _depth = 28, Min _samples _split = 34, Min _samples _leaf = 14, Criterion = gini | | SMOTE+ ENN | Algorithm = SAMME.R, n_ estimators = 178, learning_ rate = 1.5 |
| SVM | Actual | C = 0.8, gamma = 0.01, degree = 3,kernel = rbf | XG Boost | Actual | n_ estimators = 136, learning_ rate = 0.05, max_ depth = 10, min_ samples_ split = 38, max_ features = log2 |
| | SMOTE | C = 10, gamma = 0.1, degree = 4, kernel = rbf | | SMOTE | n_ estimators = 115, learning_ rate = 0.05, max_ depth = 26, min_ samples_ split = 94, max_ features = sqrt |
| | ADASYN | C = 0.1, gamma = 0.1, degree = 2, kernel = poly | | ADASYN | n_ estimators = 178, learning_ rate = 0.05, max_ depth = 4, min_ samples_ split = 90, max_ features = auto |
| | TomekLinks | C = 0.8, gamma = 0.1, degree = 3, kernel = poly | | TomekLinks | n_ estimators = 52, learning_ rate = 0.05, max_ depth = 2, min_ samples_ split = 18, max_ features = log2 |
| | ENN | C = 0.8, gamma = 0.01 degree = 4,kernel = poly | | ENN | n_ estimators = 157, learning_ rate = 0.05, max_ depth = 24, min_ samples_ split = 72, max_ features = auto |
| | SMOTE+ TomekLinks | C = 0.1,gamma = 1, degree = 4,kernel = poly | | SMOTE+ TomekLinks | n_ estimators = 73, learning_ rate = 0.5, max_ depth = 2, min_ samples_ split = 16, max_ features = sqrt |
| | SMOTE+ ENN | C = 10,gamma = 0.1, degree = 4,kernel = rbf | | SMOTE+ ENN | n_ estimators = 178, learning_ rate = 0.05, max_ depth = 26, min_ samples_ split = 68, max_ features = auto |

*(Continued)*

**Table 2.** (Continued)

| ML model | Sampling | Parameter | ML model | Sampling | Parameter |
|---|---|---|---|---|---|
| NN | Actual | Activation = identity, Hidden _layer _sizes = (50,100), solver = adam, learning _rate = constant | Light GBM | Actual | n_ estimators = 200, learning_ rate = 0.05, max_ depth = 4 |
| | SMOTE | activation = logistic, hidden _layer _sizes= (50,100), solver = adam, learning _rate = constant | | SMOTE | n_ estimators = 94, learning_ rate = 0.05, max_ depth = 28 |
| | ADASYN | activation = relu, hidden _layer _sizes= (100,), solver = adam, learning _rate = constant | | ADASYN | n_ estimators = 94, learning_ rate = 0.05, max_ depth = 28 |
| | TomekLinks | activation = identity, hidden _layer _sizes= (50,100), solver = lbfgs, learning _rate = invscaling | | TomekLinks | n_ estimators = 178, learning_ rate = 0.005, max_ depth = 14 |
| | ENN | activation = logistic, hidden _layer _sizes= (100,), solver = adam, learning _rate = adaptive | | ENN | n_ estimators = 178, learning_ rate = 0.05, max_ depth = 8 |
| | SMOTE+ TomekLinks | activation = relu, hidden_layer_sizes= (100,), solver = adam, learning_rate = constant | | SMOTE+ TomekLinks | n_ estimators = 94, learning_ rate = 0.05, max_ depth = 16 |
| | SMOTE+ ENN | activation = tanh, hidden_layer_sizes= (100,), solver = adam, learning_rate = invscaling | | SMOTE+ ENN | n_ estimators = 157, learning_ rate = 0.5, max_ depth = 20 |
| KNN | Actual | n_neighbors = 15, weights = distance, algorithm = brute, metric = manhattan | CatBoost | Actual | Depth = 6, learning_ rate = 0.05, iterations = 70, l2_leaf_reg = 9 |
| | SMOTE | n_neighbors = 9, weights = distance, algorithm = kd_tree, metric = manhattan | | SMOTE | Depth = 8, learning_ rate = 0.05, iterations = 60, l2_leaf_reg = 3 |
| | ADASYN | n_neighbors = 5, weights = distance, algorithm = auto, metric = manhattan | | ADASYN | Depth = 10, learning_ rate = 0.05, iterations = 60, l2_leaf_reg = 5 |
| | TomekLinks | n_neighbors = 15, weights = distance, algorithm = kd_tree, metric = manhattan | | TomekLinks | Depth = 6, learning_ rate = 0.05, iterations = 80, l2_leaf_reg = 1 |
| | ENN | n_neighbors = 7, weights = distance, algorithm = ball_tree, metric = manhattan | | ENN | Depth = 10, learning_ rate = 0.05, iterations = 80, l2_leaf_reg = 3 |
| | SMOTE+ TomekLinks | n_neighbors = 5, weights = distance, algorithm = auto, metric = manhattan | | SMOTE+ TomekLinks | Depth = 10, learning_ rate = 0.5, iterations = 50, l2_leaf_reg = 9 |

*(Continued)*

**Table 2.** (Continued)

| ML model | Sampling | Parameter | ML model | Sampling | Parameter |
|---|---|---|---|---|---|
| | SMOTE+ ENN | n_neighbors=7, weights=distance, algorithm=auto, metric=manhattan | | SMOTE+ ENN | Depth=10, learning_ rate=0.5, iterations=60, l2_leaf_reg=1 |
| Random Forest | Actual | n_estimators=178, max_features=sqrt, max_depth=20, min_samples_leaf=6, min_samples_split=76 | GBM | Actual | n_ estimators=94, learning_ rate=0.05, max_ depth=4, max_ features=log2 |
| | SMOTE | n_estimators=73, max_features=sqrt, max_depth=28, min_samples_leaf=4, min_samples_split=8 | | SMOTE | n_ estimators=157, learning_ rate=0.005, max_ depth=22, max_ features=log2 |
| | ADASYN | n_estimators=136, max_features=log2, max_depth=18, min_samples_leaf=6, min_samples_split=42 | | ADASYN | n_ estimators=178, learning_ rate=0.05, max_ depth=16, max_ features=log2 |
| | TomekLinks | n_estimators=157, max_features=auto, max_depth=24, min_samples_leaf=2, min_samples_split=32 | | TomekLinks | n_ estimators=73, learning_ rate=0.05, max_ depth=2, max_ features=sqrt |
| | ENN | n_estimators=115, max_features=log2, max_depth=30, min_samples_leaf=6, min_samples_split=4 | | ENN | n_ estimators=136, learning_ rate=0.05, max_ depth=20, max_ features=log2 |
| | SMOTE+ TomekLinks | n_estimators=94, max_features=auto, max_depth=14, min_samples_leaf=2, min_samples_split=34 | | SMOTE+ TomekLinks | n_ estimators=200, learning_ rate=0.05, max_ depth=16, max_ features=sqrt |
| | SMOTE+ ENN | n_estimators=157, max_features=sqrt, max_depth=20, min_samples_leaf=4, min_samples_split=34 | | SMOTE+ ENN | n_ estimators=178, learning_ rate=0.05, max_ depth=24, max_ features=log2 |

## Internal overfitting diagnostics

Repeated stratified five-fold CV yielded an average F1 of 0.934±0.012; nested CV showed an outer-fold mean F1 of 0.965±0.010; the held-out test F1 was 0.9447 (Supplementary Table S3A in S4 Appendix). Learning-curve behavior showed convergence between training and validation performance as sample size increased, supporting strong generalization while acknowledging that optimism is possible without external validation (S1 Fig).

## Test-set performance with bootstrap confidence intervals

On the independent test set (n=672), ExtraTrees+SMOTE+ENN achieved Precision 0.92 (95% CI 0.90–0.94), Recall 0.95 (0.93–0.97), F1 0.94 (0.92–0.96), Accuracy 0.91 (0.89–0.94), Specificity 0.83 (0.80–0.87), ROC-AUC 0.95

**Table 3. Performance of the machine learning algorithms.**

| Model | Class balancing technique | Precision | Recall | F1-score | AUC-ROC | AUC-PR | Cohens-kappa | G-mean | Matthews correlation coefficient | Accu-racy | Specifi-city |
|---|---|---|---|---|---|---|---|---|---|---|---|
| Logistic Regression | actual | 0.57 | 0.21 | 0.31 | 0.71 | 0.48 | 0.21 | 0.45 | 0.24 | 0.78 | 0.95 |
| Logistic Regression | smote | 0.37 | 0.62 | 0.46 | 0.72 | 0.54 | 0.24 | 0.65 | 0.26 | 0.67 | 0.68 |
| Logistic Regression | adasyn | 0.35 | 0.65 | 0.46 | 0.71 | 0.54 | 0.22 | 0.65 | 0.25 | 0.64 | 0.64 |
| Logistic Regression | tomeklink | 0.36 | 0.67 | 0.47 | 0.72 | 0.55 | 0.24 | 0.66 | 0.27 | 0.65 | 0.64 |
| Logistic Regression | enn | 0.38 | 0.57 | 0.45 | 0.71 | 0.52 | 0.24 | 0.64 | 0.25 | 0.68 | 0.72 |
| Logistic Regression | smote+tomek | 0.34 | 0.62 | 0.44 | 0.71 | 0.53 | 0.21 | 0.63 | 0.23 | 0.64 | 0.65 |
| Logistic Regression | smote+enn | 0.30 | 0.80 | 0.44 | 0.70 | 0.57 | 0.16 | 0.60 | 0.22 | 0.53 | 0.45 |
| Decision Tree | actual | 0.55 | 0.26 | 0.35 | 0.70 | 0.49 | 0.24 | 0.49 | 0.26 | 0.78 | 0.94 |
| Decision Tree | smote | 0.40 | 0.34 | 0.37 | 0.63 | 0.45 | 0.20 | 0.54 | 0.20 | 0.73 | 0.85 |
| Decision Tree | adasyn | 0.34 | 0.31 | 0.32 | 0.61 | 0.40 | 0.13 | 0.50 | 0.13 | 0.70 | 0.82 |
| Decision Tree | tomeklink | 0.49 | 0.35 | 0.41 | 0.69 | 0.50 | 0.27 | 0.56 | 0.27 | 0.77 | 0.89 |
| Decision Tree | enn | 0.37 | 0.56 | 0.45 | 0.70 | 0.52 | 0.23 | 0.63 | 0.24 | 0.68 | 0.72 |
| Decision Tree | smote+tomek | 0.30 | 0.27 | 0.28 | 0.60 | 0.36 | 0.08 | 0.46 | 0.08 | 0.69 | 0.81 |
| Decision Tree | smote+enn | 0.34 | 0.72 | 0.46 | 0.69 | 0.56 | 0.21 | 0.64 | 0.25 | 0.61 | 0.57 |
| Neural network | actual | 0.53 | 0.19 | 0.28 | 0.71 | 0.45 | 0.18 | 0.42 | 0.21 | 0.77 | 0.95 |
| Neural network | smote | 0.30 | 0.31 | 0.31 | 0.59 | 0.38 | 0.09 | 0.49 | 0.09 | 0.67 | 0.78 |
| Neural network | adasyn | 0.30 | 0.29 | 0.29 | 0.60 | 0.38 | 0.09 | 0.48 | 0.09 | 0.68 | 0.80 |
| Neural network | tomeklink | 0.54 | 0.26 | 0.35 | 0.71 | 0.48 | 0.24 | 0.49 | 0.26 | 0.78 | 0.93 |
| Neural network | enn | 0.37 | 0.59 | 0.45 | 0.70 | 0.53 | 0.24 | 0.64 | 0.25 | 0.67 | 0.70 |
| Neural network | smote+tomek | 0.32 | 0.32 | 0.32 | 0.61 | 0.40 | 0.12 | 0.51 | 0.12 | 0.69 | 0.80 |
| Neural network | smote+enn | 0.33 | 0.71 | 0.45 | 0.69 | 0.55 | 0.20 | 0.63 | 0.23 | 0.60 | 0.56 |
| Knn | actual | 0.45 | 0.18 | 0.26 | 0.68 | 0.41 | 0.15 | 0.41 | 0.17 | 0.76 | 0.94 |
| Knn | smote | 0.33 | 0.44 | 0.38 | 0.66 | 0.45 | 0.15 | 0.57 | 0.16 | 0.66 | 0.73 |
| Knn | adasyn | 0.33 | 0.42 | 0.37 | 0.64 | 0.44 | 0.15 | 0.56 | 0.15 | 0.67 | 0.74 |
| Knn | tomeklink | 0.43 | 0.20 | 0.28 | 0.68 | 0.41 | 0.15 | 0.43 | 0.17 | 0.75 | 0.92 |
| Knn | enn | 0.36 | 0.58 | 0.44 | 0.67 | 0.52 | 0.22 | 0.63 | 0.23 | 0.66 | 0.69 |
| Knn | smote+tomek | 0.33 | 0.41 | 0.36 | 0.64 | 0.44 | 0.15 | 0.55 | 0.15 | 0.67 | 0.75 |
| Knn | smote+enn | 0.32 | 0.76 | 0.45 | 0.67 | 0.56 | 0.18 | 0.62 | 0.23 | 0.57 | 0.51 |
| Random Forest | actual | 0.72 | 0.07 | 0.12 | 0.71 | 0.50 | 0.09 | 0.26 | 0.17 | 0.78 | 0.99 |
| Random Forest | smote | 0.45 | 0.33 | 0.38 | 0.69 | 0.47 | 0.23 | 0.54 | 0.24 | 0.75 | 0.88 |
| Random Forest | adasyn | 0.42 | 0.42 | 0.42 | 0.71 | 0.49 | 0.25 | 0.59 | 0.25 | 0.73 | 0.82 |
| Random Forest | tomeklink | 0.56 | 0.18 | 0.28 | 0.70 | 0.47 | 0.18 | 0.42 | 0.22 | 0.78 | 0.96 |
| Random Forest | enn | 0.37 | 0.58 | 0.46 | 0.71 | 0.53 | 0.24 | 0.64 | 0.25 | 0.68 | 0.71 |
| Random Forest | smote+tomek | 0.42 | 0.38 | 0.40 | 0.71 | 0.47 | 0.23 | 0.57 | 0.23 | 0.73 | 0.84 |
| Random Forest | smote+enn | 0.34 | 0.73 | 0.47 | 0.71 | 0.57 | 0.22 | 0.65 | 0.26 | 0.61 | 0.58 |
| Extra tree | actual | 0.80 | 0.04 | 0.08 | 0.71 | 0.53 | 0.06 | 0.20 | 0.15 | 0.78 | 1.00 |
| Extra tree | smote | 0.39 | 0.52 | 0.45 | 0.70 | 0.51 | 0.25 | 0.63 | 0.25 | 0.71 | 0.76 |
| Extra tree | adasyn | 0.41 | 0.40 | 0.40 | 0.70 | 0.47 | 0.23 | 0.58 | 0.23 | 0.73 | 0.82 |
| Extra tree | tomeklink | 0.57 | 0.20 | 0.29 | 0.70 | 0.47 | 0.20 | 0.44 | 0.24 | 0.78 | 0.95 |
| Extra tree | enn | 0.35 | 0.57 | 0.43 | 0.70 | 0.51 | 0.21 | 0.62 | 0.22 | 0.66 | 0.69 |
| Extra tree | smote+tomek | 0.42 | 0.47 | 0.44 | 0.70 | 0.50 | 0.26 | 0.61 | 0.26 | 0.73 | 0.80 |
| Extra tree | smote+enn | 0.92 | 0.95 | 0.94 | 0.95 | 0.95 | 0.79 | 0.89 | 0.79 | 0.91 | 0.83 |
| Adaboost | actual | 0.57 | 0.18 | 0.28 | 0.71 | 0.47 | 0.19 | 0.42 | 0.23 | 0.78 | 0.96 |
| Adaboost | smote | 0.50 | 0.33 | 0.40 | 0.71 | 0.49 | 0.27 | 0.55 | 0.27 | 0.77 | 0.90 |

*(Continued)*

| Model | Class balancing technique | Precision | Recall | F1-score | AUC-ROC | AUC-PR | Cohens-kappa | G-mean | Matthews correlation coefficient | Accu-racy | Speci-ficity |
|---|---|---|---|---|---|---|---|---|---|---|---|
| Adaboost | adasyn | 0.48 | 0.22 | 0.30 | 0.70 | 0.44 | 0.18 | 0.45 | 0.21 | 0.77 | 0.93 |
| Adaboost | tomeklink | 0.55 | 0.28 | 0.37 | 0.71 | 0.49 | 0.25 | 0.51 | 0.27 | 0.78 | 0.93 |
| Adaboost | enn | 0.38 | 0.58 | 0.46 | 0.71 | 0.53 | 0.24 | 0.64 | 0.26 | 0.68 | 0.71 |
| Adaboost | smote+tomek | 0.52 | 0.23 | 0.32 | 0.70 | 0.46 | 0.21 | 0.47 | 0.23 | 0.77 | 0.93 |
| Adaboost | smote+enn | 0.35 | 0.62 | 0.45 | 0.71 | 0.53 | 0.22 | 0.64 | 0.24 | 0.65 | 0.66 |
| XG Boost | actual | 0.43 | 0.25 | 0.32 | 0.66 | 0.43 | 0.18 | 0.47 | 0.19 | 0.75 | 0.90 |
| XG Boost | smote | 0.43 | 0.29 | 0.35 | 0.65 | 0.44 | 0.20 | 0.51 | 0.20 | 0.75 | 0.88 |
| XG Boost | adasyn | 0.54 | 0.29 | 0.37 | 0.71 | 0.50 | 0.26 | 0.51 | 0.28 | 0.78 | 0.93 |
| XG Boost | tomeklink | 0.52 | 0.09 | 0.15 | 0.71 | 0.41 | 0.09 | 0.29 | 0.14 | 0.77 | 0.98 |
| XG Boost | enn | 0.34 | 0.60 | 0.44 | 0.67 | 0.52 | 0.20 | 0.63 | 0.22 | 0.64 | 0.65 |
| XG Boost | smote+tomek | 0.55 | 0.26 | 0.35 | 0.70 | 0.49 | 0.24 | 0.49 | 0.27 | 0.78 | 0.94 |
| XG Boost | smote+enn | 0.35 | 0.64 | 0.45 | 0.70 | 0.54 | 0.22 | 0.64 | 0.24 | 0.64 | 0.64 |
| LGBM | actual | 0.56 | 0.25 | 0.35 | 0.70 | 0.49 | 0.24 | 0.48 | 0.26 | 0.78 | 0.94 |
| LGBM | smote | 0.48 | 0.30 | 0.37 | 0.70 | 0.47 | 0.23 | 0.52 | 0.24 | 0.76 | 0.90 |
| LGBM | adasyn | 0.47 | 0.28 | 0.35 | 0.70 | 0.46 | 0.21 | 0.50 | 0.22 | 0.76 | 0.91 |
| LGBM | tomeklink | 0.50 | 0.03 | 0.06 | 0.71 | 0.38 | 0.03 | 0.17 | 0.08 | 0.77 | 0.99 |
| LGBM | enn | 0.34 | 0.56 | 0.42 | 0.68 | 0.50 | 0.19 | 0.61 | 0.20 | 0.65 | 0.67 |
| LGBM | smote+tomek | 0.48 | 0.27 | 0.35 | 0.70 | 0.46 | 0.22 | 0.50 | 0.23 | 0.76 | 0.91 |
| LGBM | smote+enn | 0.34 | 0.63 | 0.45 | 0.69 | 0.53 | 0.21 | 0.64 | 0.23 | 0.64 | 0.64 |
| CatBoost | actual | 0.58 | 0.11 | 0.18 | 0.71 | 0.45 | 0.12 | 0.32 | 0.18 | 0.78 | 0.98 |
| CatBoost | smote | 0.46 | 0.37 | 0.41 | 0.71 | 0.49 | 0.26 | 0.57 | 0.26 | 0.75 | 0.87 |
| CatBoost | adasyn | 0.48 | 0.34 | 0.40 | 0.71 | 0.49 | 0.26 | 0.55 | 0.26 | 0.76 | 0.89 |
| CatBoost | tomeklink | 0.59 | 0.21 | 0.31 | 0.70 | 0.49 | 0.21 | 0.45 | 0.25 | 0.78 | 0.96 |
| CatBoost | enn | 0.38 | 0.61 | 0.47 | 0.71 | 0.54 | 0.26 | 0.66 | 0.27 | 0.68 | 0.70 |
| CatBoost | smote+tomek | 0.40 | 0.29 | 0.34 | 0.66 | 0.43 | 0.18 | 0.50 | 0.18 | 0.74 | 0.87 |
| CatBoost | smote+enn | 0.35 | 0.67 | 0.46 | 0.70 | 0.55 | 0.22 | 0.65 | 0.25 | 0.63 | 0.62 |
| GBM | actual | 0.63 | 0.17 | 0.27 | 0.71 | 0.50 | 0.19 | 0.41 | 0.25 | 0.79 | 0.97 |
| GBM | smote | 0.46 | 0.25 | 0.32 | 0.68 | 0.44 | 0.19 | 0.48 | 0.21 | 0.76 | 0.91 |
| GBM | adasyn | 0.47 | 0.24 | 0.32 | 0.69 | 0.44 | 0.19 | 0.47 | 0.20 | 0.76 | 0.92 |
| GBM | tomeklink | 0.58 | 0.13 | 0.21 | 0.71 | 0.45 | 0.14 | 0.35 | 0.19 | 0.78 | 0.97 |
| GBM | enn | 0.36 | 0.59 | 0.45 | 0.69 | 0.52 | 0.22 | 0.63 | 0.24 | 0.66 | 0.68 |
| GBM | smote+tomek | 0.46 | 0.26 | 0.33 | 0.69 | 0.45 | 0.20 | 0.49 | 0.21 | 0.76 | 0.91 |
| GBM | smote+enn | 0.32 | 0.76 | 0.45 | 0.69 | 0.57 | 0.19 | 0.63 | 0.23 | 0.58 | 0.52 |
| SVM | actual | 0.68 | 0.07 | 0.12 | 0.67 | 0.48 | 0.08 | 0.26 | 0.16 | 0.78 | 0.99 |
| SVM | smote | 0.35 | 0.04 | 0.07 | 0.64 | 0.30 | 0.03 | 0.20 | 0.05 | 0.76 | 0.98 |
| SVM | adasyn | 0.31 | 0.45 | 0.37 | 0.62 | 0.44 | 0.13 | 0.56 | 0.13 | 0.64 | 0.70 |
| SVM | tomeklink | 0.35 | 0.37 | 0.36 | 0.62 | 0.44 | 0.17 | 0.54 | 0.17 | 0.70 | 0.80 |
| SVM | enn | 0.41 | 0.08 | 0.13 | 0.67 | 0.35 | 0.06 | 0.27 | 0.09 | 0.76 | 0.97 |
| SVM | smote+tomek | 0.34 | 0.37 | 0.35 | 0.61 | 0.43 | 0.15 | 0.54 | 0.15 | 0.69 | 0.79 |
| SVM | smote+enn | 0.31 | 0.34 | 0.32 | 0.63 | 0.40 | 0.11 | 0.51 | 0.11 | 0.67 | 0.77 |

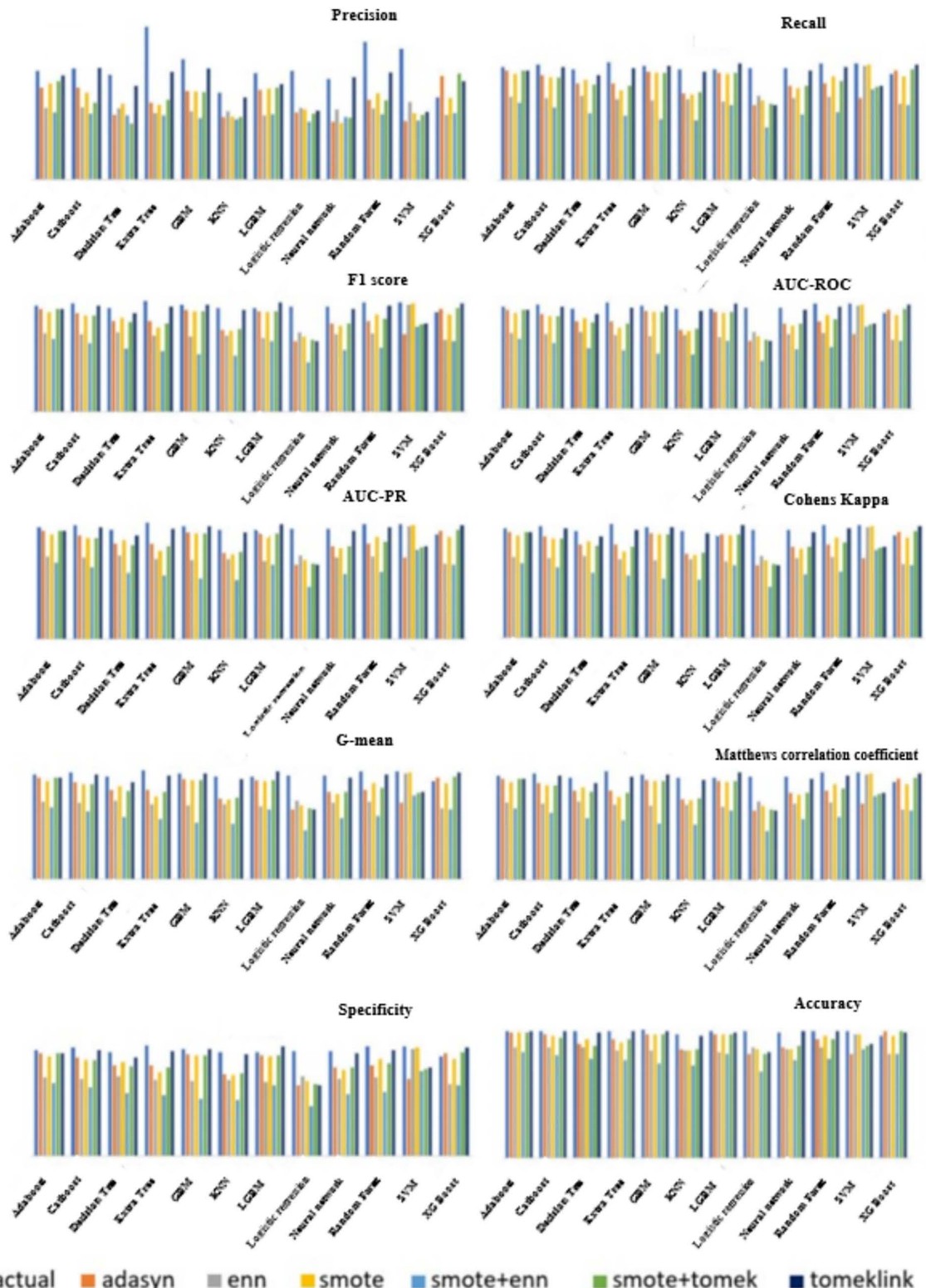

**Fig 3. Performance of the machine learning algorithms.**

**Table 4. 2×2 confusion matrix of ExtraTrees model predictions integrated with original sample.**

|  | Predicted Negative | Predicted Positive |
|---|---|---|
| Actual Negative | 518 | 2 |
| Actual Positive | 146 | 6 |

**Table 5. 2×2 confusion matrix of ExtraTrees model predictions integrated with SMOTEENN class balancing technique.**

|  | Predicted Negative | Predicted Positive |
|---|---|---|
| Actual Negative | 185 | 27 |
| Actual Positive | 20 | 440 |

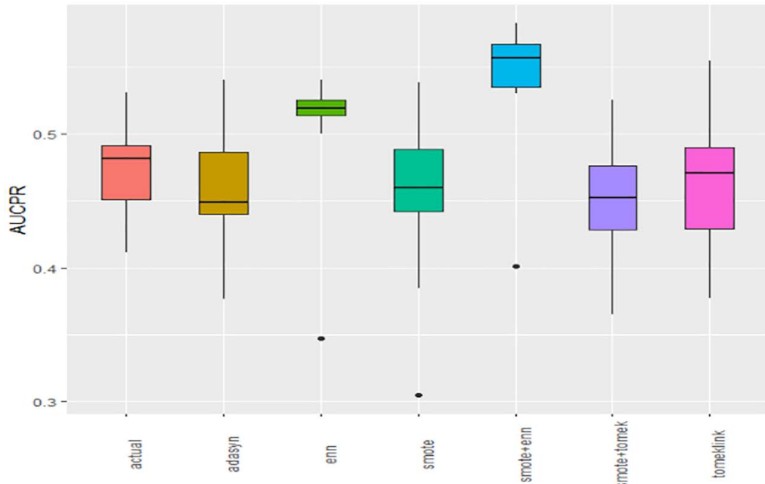

**Fig 4. The Boxplot of the original data and class balancing techniques based on AUC-PR.**

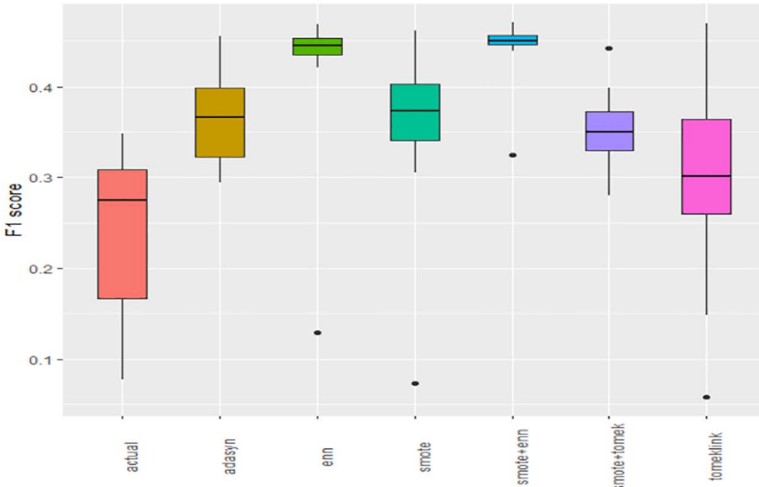

**Fig 5. The Boxplot of the original data and class balancing techniques based on F1 scores.**

**Fig 6. The Boxplot of the original data and class balancing techniques based on Recall.**

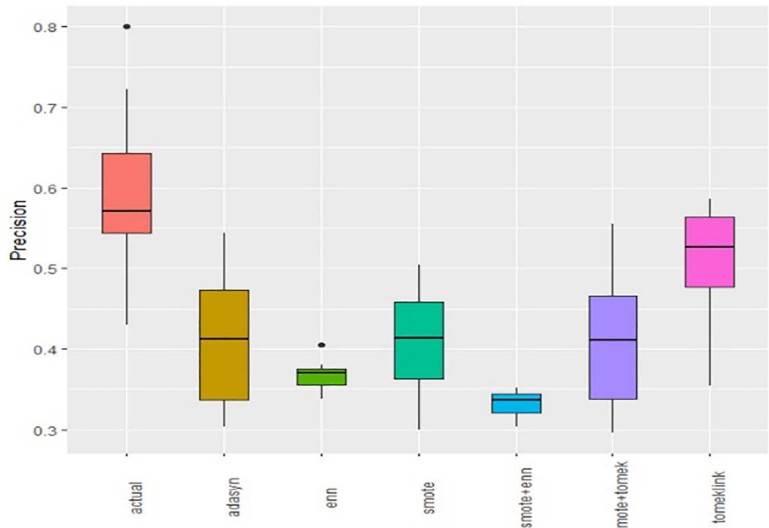

**Fig 7. The Boxplot of the original data and class balancing techniques based on Precision.**

(0.93–0.97), PR-AUC 0.95 (0.93–0.97), Cohen's κ 0.79 (0.75–0.83), MCC 0.79 (0.75–0.83) and G-mean 0.89 (0.86–0.92) (Table 6).

## Multicollinearity

Diagnostics indicated structurally inflated VIFs under one-hot encoding. Correlation screening revealed expected dependencies (e.g., respondent age < 35 vs. husband age ≥ 40, ρ = −0.82; mutually exclusive recent-birth categories, ρ = −0.89), along with dependencies among BMI categories, parity, and living-children counts. These reflect structural collinearities inherent to categorical coding. Tree-based models such as ExtraTrees are robust to these correlations and predictive performance was unaffected (Supplementary Table S5 in S4 Appendix).

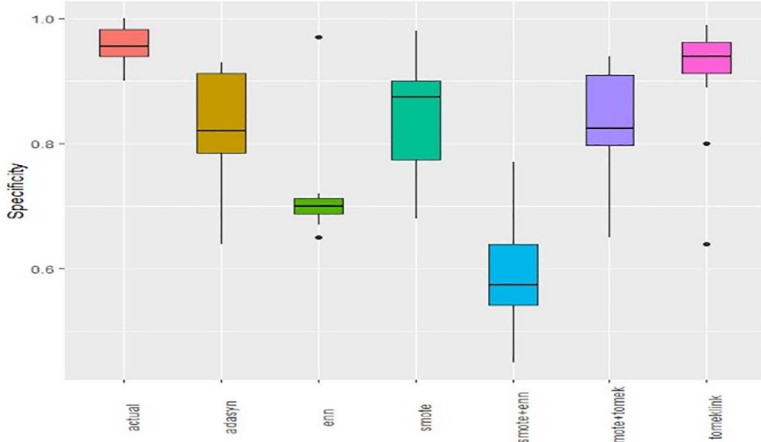

**Fig 8. The Boxplot of the original data and class balancing techniques based on Specificity.**

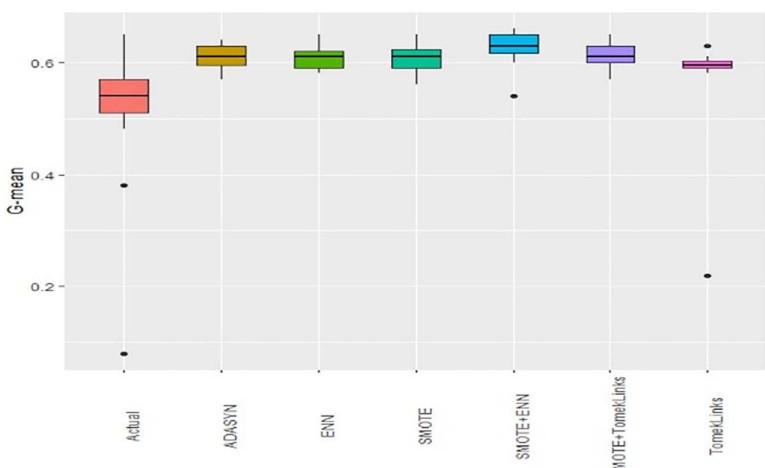

**Fig 9. The Boxplot of the original data and class balancing techniques based on G-mean.**

## Sample-flow transparency and class balance across stages

A CONSORT-style accounting of the analytic sample is shown in Supplementary Table S4A in S4 Appendix. No rows were lost to missingness; subsequent splits yielded the training and test sets used for modeling. Stratified 5-fold CV preserved the original training prevalence (23.1% hypertensive) in each fold (Supplementary Table S4B in S4 Appendix). Class balancing (e.g., SMOTE+ENN) was applied inside the training portion of each fold only; the final ExtraTrees model was refit on the full training set using SMOTE+ENN, yielding a minority-dominant training balance (Supplementary Table S4C in S4 Appendix). The independent test set (n=672) remained untouched.

## Calibration and thresholding

The ExtraTrees model was well calibrated (Brier score=0.0626; S2 Fig). The Youden-optimal decision threshold was 0.55, at which Youden's J was 0.859 (Supplementary Table S3B in S4 Appendix). Unless noted, confusion matrices and test-set metrics were computed at threshold 0.55.

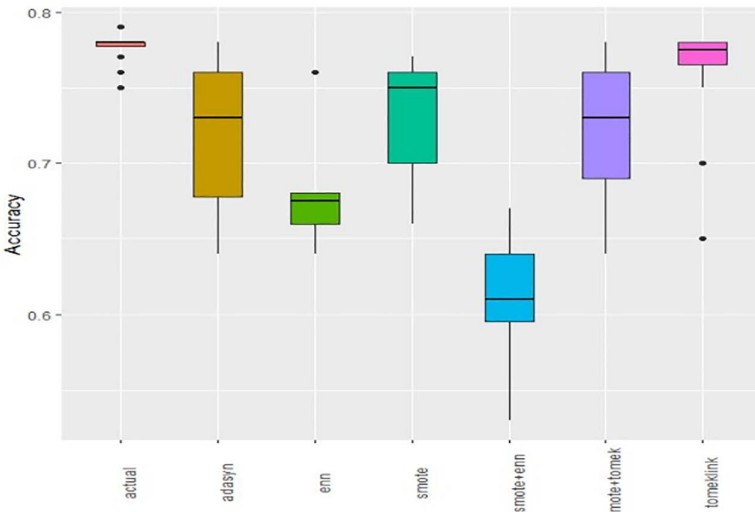

**Fig 10. The Boxplot of the original data and class balancing techniques based on Accuracy.**

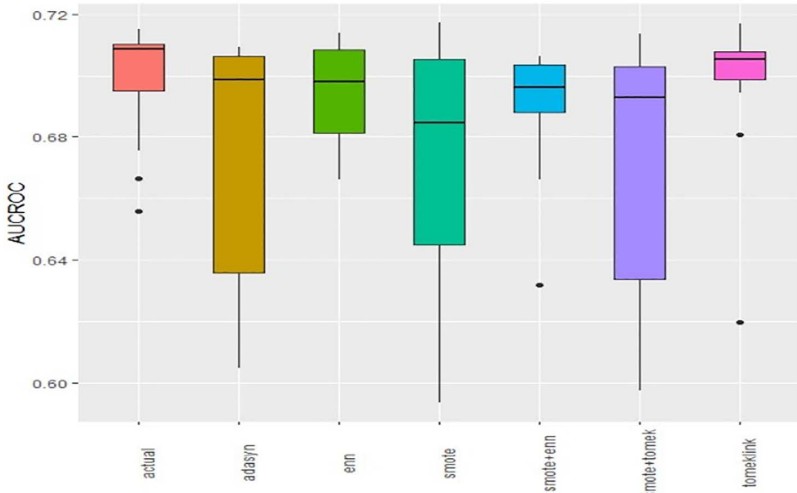

**Fig 11. The Boxplot of the original data and class balancing techniques based on AUCROC.**

Threshold consistency: Unless noted, the confusion matrix and all bootstrap test-set metrics were computed at the 0.55 decision threshold.

## Fairness/ stratified performance

Subgroup analyses were generally strong, though some divisions showed slightly lower Accuracy and F1 despite perfect discrimination (ROC-AUC = 1.0). For example, in Rajshahi, Accuracy and F1 were modestly reduced relative to overall performance (Supplementary Table S6 in S4 Appendix).

## Discussion

This study systematically evaluated 12 machine-learning (ML) algorithms with six class balancing strategies, integrating SHAP (SHapley Additive exPlanations) to identify factors associated with hypertension among married women in Bangladesh. Extra Trees combined with SMOTE+ENN (Synthetic Minority Over-sampling Technique + Edited Nearest Neighbors) achieved the best performance (F1 = 0.94; AUC-PR = 0.95). Gains were largest for recall, F1, G-mean and AUC-PR metrics better suited to imbalanced data while precision, specificity, accuracy and AUC-ROC were comparatively stable across samplers (Figs 4–11).

Our findings align with global work yet add context-specific insights. An Ethiopian stacking/XGBoost model reported slightly higher performance (F1 = 96.5%, AUC = 0.97) on a much smaller sample (n = 612) emphasizing clinical/lifestyle predictors [97]. In contrast, our socially grounded features parity, spousal education, contraceptive use performed strongly in a nationally representative cohort. Similarly, studies from Malaysia, South Korea, Japan and Norway reported AUCs ≈ 0.78–0.87 using largely clinical markers, whereas our sociodemographic model reached AUC-ROC = 0.95 [37,98–100]. Innovative approaches using wearable ECG (AUC = 0.83) or echocardiography (AUC = 0.87) show promise but have limited population-level generalizability [101,102]. A quantum-enhanced ML study reported very high scores

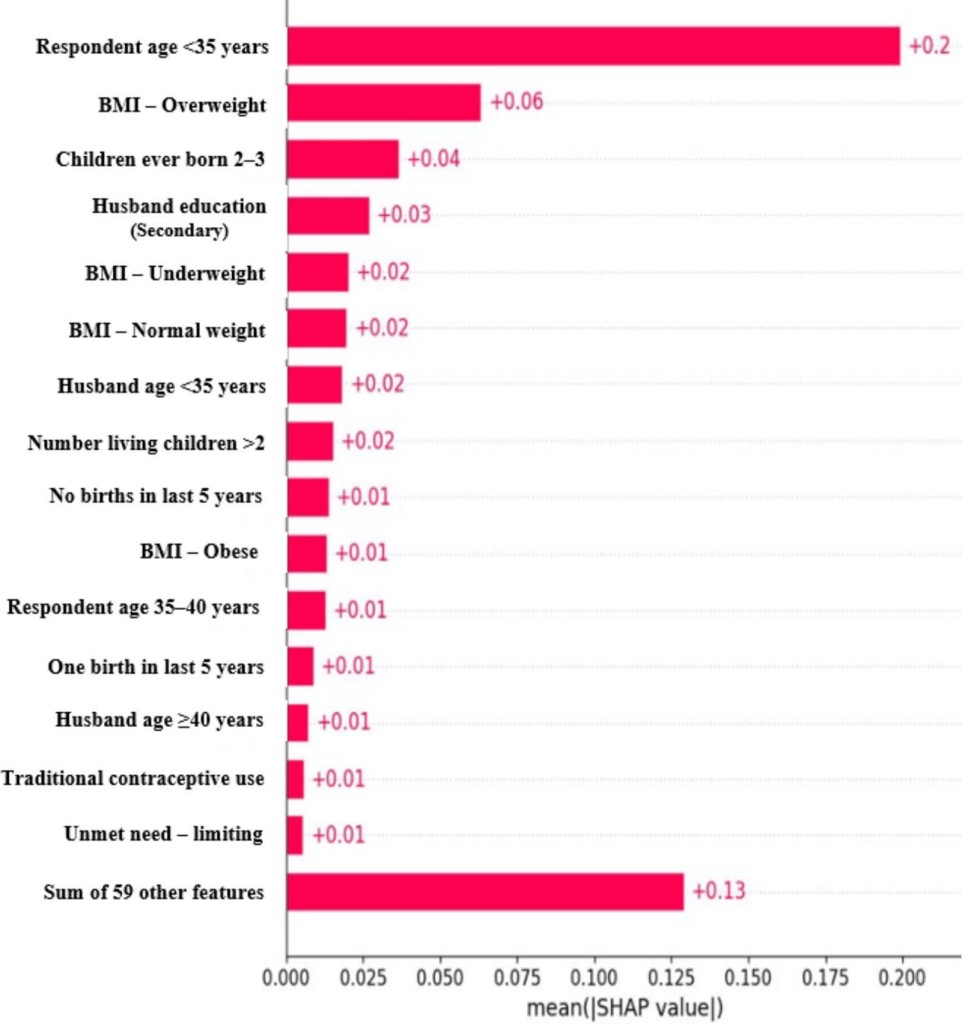

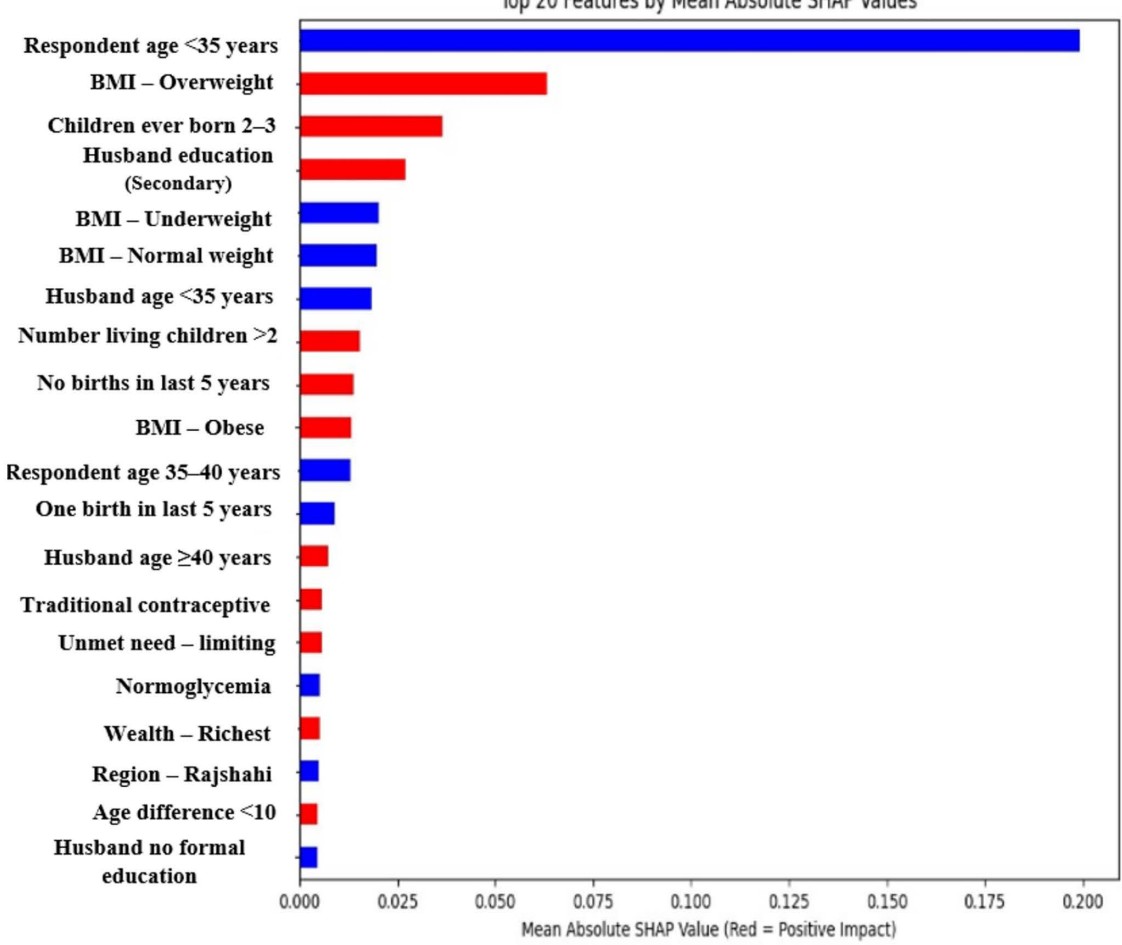

Top 20 Features by Mean Absolute SHAP Values

(F1 = 98.9%, accuracy = 98.4%), but our results underscore that carefully tuned classical ML remains highly effective and interpretable for population-specific public-health applications [103]. Quantitatively, our AUC-ROC exceeded several large clinical models by +0.08 to +0.17 absolute AUC (0.95 vs. 0.78–0.87) and our accuracy was +13 percentage points higher than Asadullah et al. (91% vs. 78%) [70].

Why emphasize PR over ROC? Precision–recall curves are more informative when the positive class is rare and accuracy can be misleadingly high due to the majority class; F1 and AUC-PR therefore provide a truer picture of minority-class performance [104–108]. Across algorithms, ensembles especially Extra Trees with SMOTE+ENN were most effective for imbalanced health data, consistent with prior reports of SMOTE+ENN paired with Random Forest, XGBoost, LightGBM, or stacked models [109–113]. The analytic sample (N = 4,253; ~23.1% hypertensive) provided sufficient positive events for supervised learning. Although several sociodemographic variables are correlated (e.g., age, parity, recent births; BMI categories), tree-based ensembles are robust to redundancy; sensitivity analyses that dropped one variable per correlated cluster left the test F1 unchanged within bootstrap uncertainty, supporting stability while preserving program-relevant interpretability.

SHAP analyses consistently highlighted age, parity, recent births, contraceptive use and spousal education as influential predictors (Figs 12a–c). Younger age (<35 years) was protective, whereas women aged 35–40 or with husbands ≥40 years showed higher risk. The positive association with husbands' secondary education counter to expectation may reflect

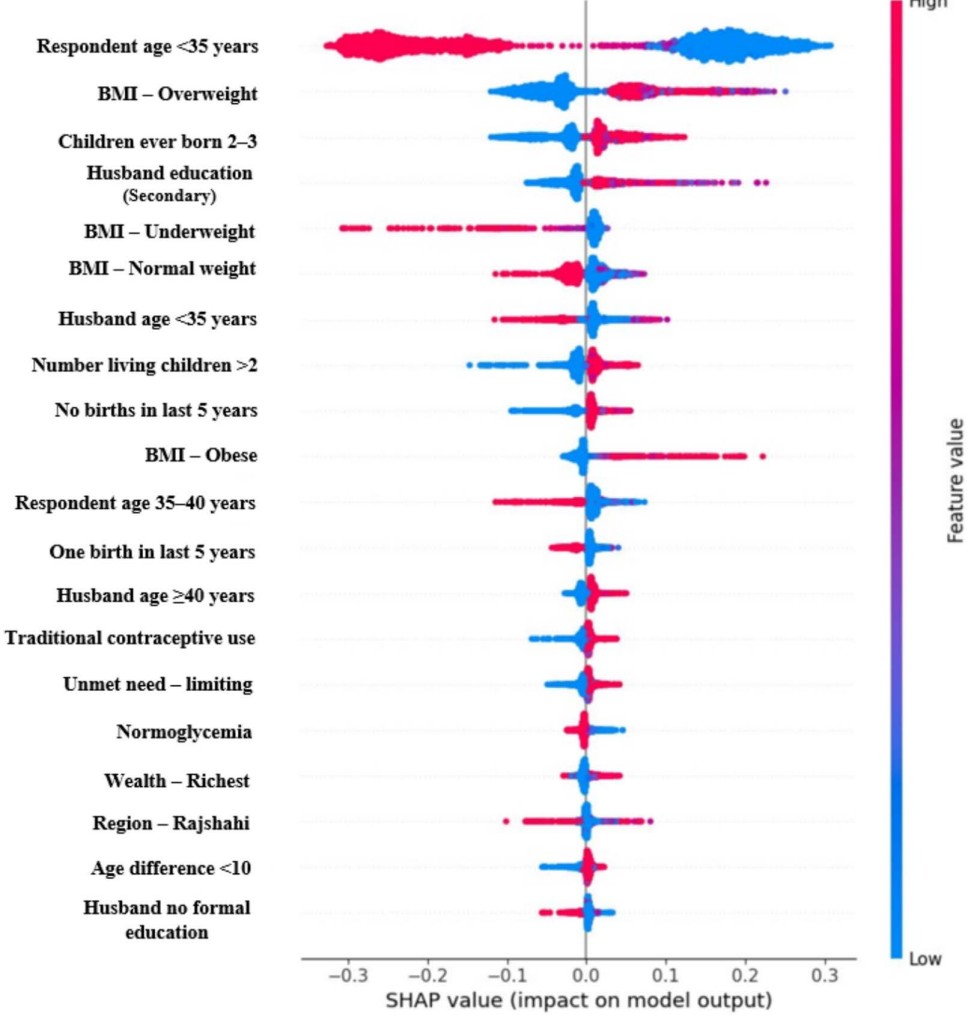

**Fig 12. a. The factors importance of hypertension among married women in Bangladesh.** b. The classification of positive and negative predictors of hypertension of married women in Bangladesh. c. The impact of the factors of married women's hypertension in Bangladesh.

socioeconomic stressors or confounding with wealth and fertility; this hypothesis warrants social-epidemiological follow-up. Our results corroborate prior Bangladesh-based work linking contraceptive use, women's age, husband's education and number of living children to hypertension risk, though earlier studies did not focus specifically on married women [114–117]

Given high recall at the Youden-optimal threshold (0.55), health systems could embed risk scoring in digital registers (e.g., eRegistries) and during family-planning or postpartum visits to triage married women for blood-pressure checks. Three actionable levers emerge: (i) integrate BP screening into contraceptive counselling and postpartum follow-up; (ii) engage husbands/partners via brief education or SMS nudges where partner age/spousal education elevate risk; and (iii) target high-parity households for home BP monitoring and referral. Geographic gradients (e.g., higher risk in Rajshahi) support district-tailored outreach. Diet-focused strategies (e.g., green coffee supplementation; DASH/Mediterranean patterns) offer complementary, population-specific interventions [118,119].

SHAP values explain model associations, not causal effects; correlated social and fertility indicators may act as proxies. Apparent positive associations (e.g., husbands' secondary education) could reflect unmeasured confounding (wealth,

**Table 6. Test-set performance of ExtraTrees with SMOTE+ENN using 1,000-replicate nonparametric bootstrap (n = 672).**

| Metric | Point estimate | 95% CI |
|---|---|---|
| Precision | 0.92 | 0.90–0.94 |
| Recall | 0.95 | 0.93–0.97 |
| F1-score | 0.94 | 0.92–0.96 |
| Accuracy | 0.91 | 0.89–0.94 |
| Specificity | 0.83 | 0.80–0.87 |
| ROC-AUC | 0.95 | 0.93–0.97 |
| PR-AUC | 0.95 | 0.93–0.97 |
| Cohen's κ | 0.79 | 0.75–0.83 |
| MCC | 0.79 | 0.75–0.83 |
| G-mean | 0.89 | 0.86–0.92 |

occupation, stress). Ethical deployment should address fairness (monitor subgroup performance gaps), transparency (document model versioning and thresholds) and potential unintended consequences (e.g., stigma or resource diversion). Thresholds should be calibrated to local prevalence and the relative costs of false negatives vs. false positives.

Internal checks repeated CV, nested CV and bootstrap Cis showed stable performance with tight uncertainty (see Results); however, training solely on BDHS-2017–18 may still inflate apparent performance. We have pre-specified external validation on BDHS-2022, applying identical preprocessing and the pre-set decision threshold (see Methods) to evaluate discrimination, calibration and transportability. This will test robustness in a later, post-COVID cohort and under any instrument changes. Findings do not generalize to unmarried women.

## Conclusion

Using BDHS 2017–18 data, we compared 12 ML algorithms and six class balancing methods for predicting hypertension among married women. Extra Trees + SMOTE+ENN was optimal (F1 = 0.94; AUC-PR = 0.95). SHAP surfaced actionable, context-specific predictors women's age, contraceptive use, parity, spousal education and household headship underscoring the value of social and demographic information when clinical data are scarce. Metrics tailored to imbalance (F1, AUC-PR) were more informative than accuracy or AUC-ROC alone.

These models can support screening and targeting in eRegistries and digital health platforms, enabling gender-sensitive, resource-aware outreach in LMICs. Policymakers can leverage findings for spousal-education initiatives, parity-focused home BP programs and region-specific strategies.

Priorities include external validation with BDHS 2022; prospective studies; integration of clinical/biometric markers; and exploration of transfer learning, cross-country adaptation and multi-omics integration. Linking models to real-time eRegistry streams could enable adaptive thresholding and continuous monitoring. Extending analyses to unmarried women and other under-represented groups will refine generalizability.

### Strengths and limitations

### Strengths

(i) First Bangladesh study (to our knowledge) to combine class-balancing and ML for hypertension prediction specifically among married women; (ii) transparent feature attribution via SHAP; (iii) rigorous evaluation with repeated CV, nested-CV, test-set bootstrapping and comparative statistics across class balancing strategies; (iv) concrete policy pathways (integration with family-planning/postpartum services, spousal engagement, district tailoring).

**Limitations**

(i) No external validation yet; BDHS may under-sample marginalized groups; findings do not extend to unmarried women; (ii) important biomedical/genetic/environmental determinants were unavailable; (iii) cross-sectional design limits causal inference SHAP explains predictions, not causes; (iv) fairness audits revealed subgroup gaps requiring monitoring and mitigation; (v) training-time class balancing alters class balance relative to the population accordingly, we calibrated thresholds on the untouched test set and reported bootstrap CIs to reduce optimism; (vi) deployment should account for gender-specific barriers in Bangladesh (mobility constraints, caregiving burden, norms around clinic attendance). Another limitation is the lack of external validation at the time of analysis, as BDHS 2022 was not released then. Since the dataset is now available, we plan to validate our deployed system using BDHS 2022, though its limited feature scope prevents full integration into model training.

## Supporting information

**S1 Appendix.  List of class balancing techniques.**
(DOCX)

**S2 Appendix.  Evaluation of machine learning algorithms.**
(DOCX)

**S3 Appendix.  Evaluation of machine learning algorithms.**
(DOCX)

**S4 Appendix.  Supplementary Tables.**
(DOCX)

**S1 Fig.  Learning curve of the ExtraTrees model for predicting hypertension among married women in Bangladesh.**
(DOCX)

**S2 Fig.  Reliability plot of the ExtraTrees model for predicting hypertension among married women in Bangladesh.**
(DOCX)

## Acknowledgments

We are grateful for the authorization to use the BDHS dataset from MEASURE DHS and the Bangladesh National Institute of Population Research and Training (NIPORT). Additionally, icddr,b extends its gratitude to the Governments of Bangladesh and Canada for their core and unrestricted support.

## Author contributions

**Conceptualization:** Novel Chandra Das, Probir Kumar Ghosh.

**Data curation:** Novel Chandra Das.

**Formal analysis:** Novel Chandra Das.

**Investigation:** Probir Kumar Ghosh, Fatema Khatun, Mohammad Ziaul Islam Chowdhury.

**Methodology:** Novel Chandra Das.

**Project administration:** Novel Chandra Das.

**Supervision:** Probir Kumar Ghosh, Fatema Khatun, Mohammad Ziaul Islam Chowdhury.

**Validation:** Mohammad Ziaul Islam Chowdhury.

**Visualization:** Novel Chandra Das.

**Writing – original draft:** Novel Chandra Das.

**Writing – review & editing:** Novel Chandra Das, Probir Kumar Ghosh, Md. Alamgir Hossain, Uddip Acharjee Shuvo, Nipa Rani Talukder, Fatema Khatun, Mohammad Ziaul Islam Chowdhury.

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
