## [Decision Letter · Decision Letter 0]

15 May 2025

Dear Dr. Das,

Thank you for submitting your manuscript to PLOS ONE. After careful consideration, we feel that it has merit but does not fully meet PLOS ONE’s publication criteria as it currently stands. Therefore, we invite you to submit a revised version of the manuscript that addresses the points raised during the review process.

We look forward to receiving your revised manuscript.

Kind regards,

Benojir Ahammed, M.Sc.

Academic Editor

PLOS ONE

**Journal Requirements:**

1. When submitting your revision, we need you to address these additional requirements. Please ensure that your manuscript meets PLOS ONE's style requirements, including those for file naming. The PLOS ONE style templates can be found at https://journals.plos.org/plosone/s/file?id=wjVg/PLOSOne_formatting_sample_main_body.pdf and https://journals.plos.org/plosone/s/file?id=ba62/PLOSOne_formatting_sample_title_authors_affiliations.pdf 2. Please note that PLOS ONE has specific guidelines on code sharing for submissions in which author-generated code underpins the findings in the manuscript. In these cases, we expect all author-generated code to be made available without restrictions upon publication of the work. Please review our guidelines at https://journals.plos.org/plosone/s/materials-and-software-sharing#loc-sharing-code and ensure that your code is shared in a way that follows best practice and facilitates reproducibility and reuse. 3. Thank you for uploading your study's underlying data set. Unfortunately, the repository you have noted in your Data Availability statement does not qualify as an acceptable data repository according to PLOS's standards. At this time, please upload the minimal data set necessary to replicate your study's findings to a stable, public repository (such as figshare or Dryad) and provide us with the relevant URLs, DOIs, or accession numbers that may be used to access these data. For a list of recommended repositories and additional information on PLOS standards for data deposition, please see https://journals.plos.org/plosone/s/recommended-repositories. 4. In the online submission form, you indicated that “The codes are based on python and R and available from the corresponding author on proper request”. All PLOS journals now require all data underlying the findings described in their manuscript to be freely available to other researchers, either a. In a public repository, b. Within the manuscript itself, or c. Uploaded as supplementary information.This policy applies to all data except where public deposition would breach compliance with the protocol approved by your research ethics board. If your data cannot be made publicly available for ethical or legal reasons (e.g., public availability would compromise patient privacy), please explain your reasons on resubmission and your exemption request will be escalated for approval. 5. Please amend either the abstract on the online submission form (via Edit Submission) or the abstract in the manuscript so that they are identical.

Reviewers' comments:

Reviewer's Responses to Questions

**Comments to the Author**

1. Is the manuscript technically sound, and do the data support the conclusions?

Reviewer #1: Partly

Reviewer #2: Yes

2. Has the statistical analysis been performed appropriately and rigorously?

Reviewer #1: Yes

Reviewer #2: Yes

3. Have the authors made all data underlying the findings in their manuscript fully available?

Reviewer #1: No

Reviewer #2: Yes

4. Is the manuscript presented in an intelligible fashion and written in standard English?

Reviewer #1: Yes

Reviewer #2: Yes

**Reviewer #1:**  Dear Authors

Summary of the Study:

The study evaluates machine learning (ML) algorithms to identify predictive factors of hypertension among married women in Bangladesh using data from the Bangladesh Demographic and Health Survey (2017–18). The authors employed 12 ML algorithms and 6 class-balancing techniques, with the Extra Tree algorithm combined with SMOTE+ENN yielding the highest performance (F1-score: 94%). SHAP analysis highlighted key factors, such as overweight status, parity, and partner’s education, as significant predictors.

Strengths of the Study:

• Methodology: The comprehensive evaluation of 12 ML algorithms and 6 class-balancing techniques, validated through parametric and non-parametric tests, strengthens the robustness of the findings.

• Interpretability: Integration of SHAP values enhances model transparency, enabling clear identification of both global and local predictive factors.

• Focus on married women in Bangladesh addresses a critical gap in hypertension research.

• Effective use of imbalanced data techniques (e.g., SMOTE+ENN) mitigates bias and improves minority-class prediction.

Weaknesses of the Study:

• The provided link does not lead to the dataset which makes it impossible to verify the findings because of insufficient reproducibility.

• The models were not externally validated, limiting generalizability.

• Insufficient rationale for selecting the Extra Tree algorithm over other high-performing models (e.g., XGBoost, Random Forest).

Discussion of Specific Areas for Improvement:

Major Issues

1. The dataset link provided (https://dhsprogram.com/data/dataset_admin/index.cfm) does not grant direct access to the BDHS 2017–18 data. The authors must clarify access procedures (e.g., registration requirements) or provide an alternative repository link

2. Quantitative results are presented in the study for the 12 tested machine learning algorithms yet a clear academic methodological approach remains inadequate. The authors need to create a single performance metrics table in the main text to display F1-score, AUC-PR, recall, precision, accuracy alongside each algorithm-class-balancing combination. A model-specific table must present all the hyperparameter settings selected in order to guarantee experimental repeatability. This enhanced approach will make results more transparent and allow results comparison while making the research methodological practices stronger..

3. The model requires external validation through independent data to validate its ability to generalize accurately. You should document unavailability of verification data while recommending researched methods for future validation experiments.

4. The author demonstrates why their Extra Trees model performed best by exploring features regarding parameter optimization and variable relationships. The research credibility can be improved by comparing results from equivalent studies which employed XGBoost in hypertension prediction.

Minor Issues

1. Standardize terms (e.g., use "class-balancing techniques" instead of variations like "class-balanced techniques").

2. Ensure figures (e.g., SHAP plots) are labeled clearly, with legible font sizes and color contrasts for accessibility.

3. Minor errors exist (e.g., "infectundity" → "infecundity," ). Perform thorough proofreading.

**Reviewer #2: ** 1. The author has not used the latest Bangladesh Demographic and Health Survey 2022 (URL: https://www.dhsprogram.com/pubs/pdf/FR386/FR386.pdf). Is there any special reason. Please respond and justify.

2. The author have used data for 4,253 married women in Bangladesh. As we are aware that to assess the predictive power of machine learning algorithms, a sufficiently large sample is required.

3. Please keep the format and use of comma (,) properly throughout the text such as Row Number 154, 214, 218, 395, 482 etc.

4. Provide heading t table 2 and avoid repeatedly mentioning A and p-values within the table.

5. Avoid repeatedly mentioning chi-squared, df and p-values in table 5.

6. The conclusion part of the study is very short. It should be explanatory and exhaustive in nature.

**Do you want your identity to be public for this peer review?** For information about this choice, including consent withdrawal, please see our Privacy Policy

Reviewer #1: **Yes: ** Ali Abbas Abbod

Reviewer #2: **Yes: ** Muhammad Abdus Salam

---

## [Author Response · Author response to Decision Letter 1]

3 Jun 2025

The Reviewers,

We are grateful for the thoughtful and constructive comments provided on our manuscript. We have carefully addressed each point raised and made the necessary revisions to improve the clarity, rigor, and reproducibility of our work. Below, we provide a detailed, point-by-point response to each of the comments.

Reviewer #1

Comment: Weaknesses of the Study:

Comment: The provided link does not lead to the dataset which makes it impossible to verify the findings because of insufficient reproducibility.

“Response: We acknowledge that the previously provided dataset link led to the general DHS dataset access portal and not directly to the BDHS 2017–18 dataset. Since BDHS data are governed by the DHS Program, direct public sharing is restricted. However, researchers can request access by creating a free account and submitting a data use request. To improve reproducibility and transparency, we have:

Updated the Data Availability Statement to include explicit access instructions for obtaining the BDHS 2017–18 dataset.

Shared all analysis code publicly on GitHub at https://github.com/NobleNovel/Hypertension_Maternal, which includes Python and R scripts for data preprocessing, model training, evaluation, and figure generation.

Comment : The models were not externally validated, limiting generalizability.

“Response: We agree that external validation is critical to assess model generalizability. Unfortunately, at the time of this study, no publicly accessible dataset with comparable features to BDHS 2017–18 was available, including the then-unreleased BDHS 2022. We have now:

Acknowledged this limitation in the Conclusion section line 613-617 (page 39).

Discussed plans to validate the model using the newly released BDHS 2022 dataset in future work.

Comment: Insufficient rationale for selecting the Extra Tree algorithm over other high-performing models (e.g., XGBoost, Random Forest).

“Response: Thank you for highlighting the need for a clearer rationale regarding model selection. While we evaluated 12 machine learning algorithms in combination with 6 different class-balancing techniques, the selection of the Extra Trees model was based on a systematic performance comparison across multiple metrics (F1-score, AUC-PR, precision, recall, accuracy) under each sampling strategy. As detailed in the section “Classification efficacy” line 365-379, page 24-25 and new performance metrics table (see lines 381, page 25-26)), Extra Trees consistently demonstrated superior performance across most evaluation criteria, particularly in handling class imbalance and generalization within the training dataset.

Comment: The dataset link provided (https://dhsprogram.com/data/dataset_admin/index.cfm) does not grant direct access to the BDHS 2017–18 data. The authors must clarify access procedures (e.g., registration requirements) or provide an alternative repository link

“Response: Thank you for pointing this out. We acknowledge that the link previously provided directs to the general dataset access portal of the DHS Program rather than directly to the BDHS 2017–18 dataset. As the BDHS 2017–18 data are owned by a third party (The DHS Program), direct public sharing is restricted. However, the dataset is freely available upon request for registered users. We have now updated the Data and Code Availability Statement to clarify that users must register for a free account on the DHS Program website and submit a data request specifying the BDHS 2017–18 dataset. Once approved, users can download the dataset for research purposes. The updated link and access instructions are included in the revised Data Availability Statement. We hope this clarification resolves the concern regarding data accessibility.”

Comment: Quantitative results are presented in the study for the 12 tested machine learning algorithms yet a clear academic methodological approach remains inadequate. The authors need to create a single performance metrics table in the main text to display F1-score, AUC-PR, recall, precision, accuracy alongside each algorithm-class-balancing combination. A model-specific table must present all the hyperparameter settings selected in order to guarantee experimental repeatability. This enhanced approach will make results more transparent and allow results comparison while making the research methodological practices stronger.

“Response: Thank you for this valuable suggestion. We agree that presenting a consolidated performance metrics table and clearly documenting hyperparameter settings will significantly improve the methodological transparency and reproducibility of our study. In response, we have created a comprehensive performance metrics table under the section “Classification efficacy”in the main text that displays the F1-score, AUC-PR, recall, precision, and accuracy for each machine learning algorithm across all class-balancing techniques used see line 365-379, page 24-25 and new performance metrics table (see lines 381, page 25-26),. This allows for straightforward comparison of model performance under different conditions. Additionally, we have included a section “Parameter optimization line 346-362, page 19-20 ” separate model-specific table that lists all hyperparameter settings selected for each algorithm, along with brief descriptions of how they were determined see line 364, page 20-24”

Comment: The model requires external validation through independent data to validate its ability to generalize accurately. You should document unavailability of verification data while recommending researched methods for future validation experiments.

“Response: We appreciate the comment regarding the need for external validation to assess the generalizability of the model. We fully acknowledge the importance of validating machine learning models using independent datasets to ensure robustness and real-world applicability. At the time this study was conducted, the BDHS 2022 dataset had not yet been released, and no publicly accessible independent datasets with comparable features to BDHS 2017–18 were available for external validation.

Looking forward, we plan to evaluate our model using the BDHS 2022 data for next version of model.”

Comment: The author demonstrates why their Extra Trees model performed best by exploring features regarding parameter optimization and variable relationships. The research credibility can be improved by comparing results from equivalent studies which employed XGBoost in hypertension prediction.

“Response: Thank you for this insightful comment. We agree that situating our findings within the context of existing research can strengthen the credibility and relevance of our study. In response, we have expanded the discussion section to include a comparison with recent studies that employed XGBoost for hypertension prediction. Specifically, we highlight performance metrics such as accuracy, F1-score, and AUC reported in those studies and compare them to the results obtained in our work. We also discuss differences in dataset characteristics, feature sets, and class balancing approaches that may account for performance variations. This comparative analysis not only reinforces the strengths of our Extra Trees model but also provides a clearer understanding of its relative performance and potential advantages in similar contexts. Please see line 514-553, page 35-37”

Comment: Standardize terms (e.g., use "class-balancing techniques" instead of variations like "class-balanced techniques").

“Response: Thank you for pointing this out. We have carefully reviewed the manuscript and standardized the terminology throughout, consistently using the term "class-balancing techniques" to ensure clarity and consistency. This change improves the readability and professionalism of the manuscript please see line 29,176,257,282,395,401,404,412,425,430,503,582,626 (page 2,8,12,13,28,29,31,34,38, and 40)”

Comment: Ensure figures (e.g., SHAP plots) are labeled clearly, with legible font sizes and color contrasts for accessibility.

“Response: Due to limitations in re-running the analysis, we have enhanced the clarity of the existing SHAP plot by adjusting the resolution, font size, and labeling using graphic editing tools. The revised figure now conforms to PLOS’s formatting guidelines (TIFF format, ≥600 DPI, RGB color mode). Additionally, we have clarified the color coding for accessibility in both the figure legend and caption.

We have also included the following comment, as requested:

“While revising your submission, please upload your figure files to the Preflight Analysis and Conversion Engine (PACE) digital diagnostic tool, https://pacev2.apexcovantage.com/. PACE helps ensure that figures meet PLOS requirements. To use PACE, you must first register as a user. Registration is free. Then, login and navigate to the UPLOAD tab, where you will find detailed instructions on how to use the tool. If you encounter any issues or have any questions when using PACE, please email PLOS at figures@plos.org. Please note that Supporting Information files do not need this step.”

Comment: Minor errors exist (e.g., "infectundity" → "infecundity," ). Perform thorough proofreading.

“Response: Thank you for pointing out these minor errors. We have carefully proofread the entire manuscript and corrected all typographical and spelling mistakes, including changing “infectundity” to “infecundity.” This thorough review has improved the overall clarity and quality of the manuscript.”

Reviewer #2

Comment: The author has not used the latest Bangladesh Demographic and Health Survey 2022 (URL: https://www.dhsprogram.com/pubs/pdf/FR386/FR386.pdf). Is there any special reason? Please respond and justify.

“Response: Thank you for bringing this to our attention. When we conducted our analysis, the most recent publicly available dataset was the Bangladesh Demographic and Health Survey 2017–18; the 2022 survey data had not yet been released or made accessible for research purposes. We acknowledge that using the 2022 data could improve the relevance and accuracy of our findings. Therefore, we plan to incorporate the 2022 dataset in the next version of our model to update and validate its performance. This will be a key focus in our future work.”

Comment: The author have used data for 4,253 married women in Bangladesh. As we are aware that to assess the predictive power of machine learning algorithms, a sufficiently large sample is required.

“Response: Thank you for this important observation. We acknowledge that sample size is a crucial factor in the training and evaluation of machine learning models. The dataset of 4,253 married women from BDHS 2017–18 represents the most complete and relevant data available for our study population, with carefully selected features and sufficient variability to support model development. Additionally, we applied robust validation techniques, including cross-validation and class-balancing methods, to enhance the reliability of our results despite the sample size.

Furthermore, Previous studies have demonstrated that good predictive performance can be achieved even with smaller sample sizes than ours.”

Comment: Please keep the format and use of comma (,) properly throughout the text such as Row Number 154, 214, 218, 395, 482 etc.

“Response: Thank you for highlighting the issues with comma usage and formatting. We have carefully reviewed the manuscript and corrected all instances to ensure proper and consistent use of commas throughout the text, including in the specified rows. This revision improves the readability and professionalism of the manuscript p-lease see line 151,211,213,215,217,218,219,220,221,222,223,224,225,226, 426 page 7,11,12,31.”

Comment: Provide heading t table 2 and avoid repeatedly mentioning A and p-values within the table.

“Response: Thank you for the helpful suggestion. We have added a clear and descriptive heading to Table 2 to improve its clarity. Additionally, we have streamlined the presentation by avoiding repetitive mentions of “A” and p-values within the table, making it easier to read and interpret. Please see line 405, page 28”

Comment: Avoid repeatedly mentioning chi-squared, df and p-values in table 5.

“Response: Thank you for the valuable suggestion. We have revised Table 5 to avoid repetitive mentions of chi-squared values, degrees of freedom (df), and p-values by consolidating these statistics where appropriate. This enhances the table’s clarity and readability without compromising the presentation of important statistical information. Please see line 420, page 29”

Comment: The conclusion part of the study is very short. It should be explanatory and exhaustive in nature.

“Response: Thank you for this valuable feedback. We have expanded the conclusion section to provide a more detailed and comprehensive summary of our key findings, their implications, and potential applications. The revised conclusion also highlights the strengths and limitations of the study, as well as recommendations for future research, making it more explanatory and exhaustive as suggested. See line 591-623 , page 38-40 ”

Comment: While revising your submission, please upload your figure files to the Preflight Analysis and Conversion Engine (PACE) digital diagnostic tool, https://pacev2.apexcovantage.com/. PACE helps ensure that figures meet PLOS requirements. To use PACE, you must first register as a user. Registration is free. Then, login and navigate to the UPLOAD tab, where you will find detailed instructions on how to use the tool. If you encounter any issues or have any questions when using PACE, please email PLOS at figures@plos.org. Please note that Supporting Information files do not need this step.

“ Response: Thank you for the guidance regarding figure preparation. We have registered with the PACE digital diagnostic tool and uploaded all figure files as instructed to ensure they meet PLOS requirements. We carefully reviewed the outputs and made any necessary adjustments to improve figure quality and compliance. We appreciate this resource for helping us enhance the clarity and presentation of our figures and uploaded the figures accordingly.”

Best regards,

Novel Chandra Das

Research officer, icddrb

---

## [Decision Letter · Decision Letter 1]

7 Sep 2025

Thank you for submitting your manuscript to PLOS ONE. After careful consideration, we feel that it has merit but does not fully meet PLOS ONE’s publication criteria as it currently stands. Therefore, we invite you to submit a revised version of the manuscript that addresses the points raised during the review process.

We look forward to receiving your revised manuscript.

Kind regards,

Benojir Ahammed, M.Sc.

Academic Editor

PLOS ONE

**Journal Requirements:**

Reviewers' comments:

Reviewer's Responses to Questions

**Comments to the Author**

Reviewer #3: All comments have been addressed

Reviewer #4: (No Response)

2. Is the manuscript technically sound, and do the data support the conclusions?

Reviewer #3: Yes

Reviewer #4: Partly

3. Has the statistical analysis been performed appropriately and rigorously?

Reviewer #3: Yes

Reviewer #4: Yes

4. Have the authors made all data underlying the findings in their manuscript fully available?

Reviewer #3: Yes

Reviewer #4: No

5. Is the manuscript presented in an intelligible fashion and written in standard English?

Reviewer #3: Yes

Reviewer #4: Yes

**Reviewer #3: ** Thank you for reviewing the article. This a good scientific paper with public health importance. The article will be more palatable and readable if authors make it simple for wider audience. 

**Reviewer #4:**  Detailed Comments

Title and Abstract

The title is clear and descriptive, effectively capturing the dual focus on evaluating ML algorithms and identifying hypertension factors. It could perhaps be tightened for conciseness, something like "Machine Learning Approaches to Predict and Rank Hypertension Risk Factors Among Married Women in Bangladesh" – but that's minor.

The abstract is generally well-structured, summarizing the introduction, methods, results, and conclusion. I like how it highlights key metrics (e.g., F1 score jumping from 8% to 94% with SMOTE+ENN) and the top positive/negative factors from SHAP. However, there are a few awkward phrasings that make it feel a bit rough – for instance, "we aimed to develop a machine learning model to identify and rank predictive factors" could be more precise as "we developed..." since it's a completed study. Also, the prevalence stats (76.9% normal BP vs. 23.1% hypertensive) are useful, but tying them to broader implications right away might help. Grammatically, "infectundity" in the methods section should be "infecundity" (noted in responses to reviewers, but check if it's fixed). Overall, it's informative, but revise for smoother flow and ensure it matches the submission form version exactly, as flagged by the journal.

Introduction

The introduction sets up the problem nicely, starting with global hypertension stats and narrowing to Bangladesh, with a focus on women and married subgroups. I appreciate the literature review on ML applications in health outcomes – it's comprehensive, covering everything from SVMs in CHD prediction to SHAP's advantages over LASSO/ANOVA. References seem up-to-date and relevant, like the emphasis on tree-based models outperforming logistic regression.

That said, it feels a tad lengthy in places, with some repetition (e.g., multiple paragraphs on ML superiority). The knowledge gap – why married women specifically? – is stated, but could be backed with more evidence on cultural/economic factors in Bangladesh. Also, since the current date is 2025, mentioning why 2017-18 data was used over the 2022 BDHS (as addressed in reviewer responses) should be integrated here for transparency. In my opinion, this section is strong but could be streamlined to about 80% of its length to keep the reader engaged.

Methods

This is one of the stronger sections, with detailed descriptions of data source (BDHS 2017-18, n=4,253), preprocessing, and analysis. The choice of 12 algorithms (e.g., Extra Trees, XGBoost, SVM) and 6 balancing techniques (SMOTE, ADASYN, etc.) is justified well for handling imbalance. Validation methods (hold-out + repeated stratified k-fold) and hyperparameter tuning via random search are appropriate, and the evaluation metrics (F1, AUC-PR, etc.) suit imbalanced data. SHAP for feature importance is a great addition for interpretability.

However, a few clarifications are needed. How were features selected initially? The manuscript mentions demographic/clinical vars, but no explicit feature engineering or selection (e.g., Boruta or LASSO, as hinted in intro). Sample size justification is weak – 4,253 is decent, but discuss power for ML, especially with imbalance. Ethical considerations are covered, but confirm if any sensitive vars (e.g., religion) were handled with care. Hardware/software details are fine, but the GitHub repo for code is a plus – ensure it's fully reproducible. I'd suggest adding a flowchart for the workflow to make it easier to follow.

Results

The results are presented logically, starting with descriptive stats (Table 1 is comprehensive, showing hypertension stratification by demographics). Prevalence of 23.1% aligns with prior studies, and breakdowns (e.g., higher in Rangpur, urban areas) add context. Model performances are detailed, with Extra Trees + SMOTE+ENN shining (F1=94%, big improvement from baseline). SHAP plots (Figs 11a-c) effectively rank factors – age <35 negative, overweight positive, etc. – and the directionality (blue/red bars) is intuitive.

But some issues: Tables 4-8 on stats tests (Anderson-Darling, Friedman) are dense; consider consolidating or moving non-essential to supp. Figures need better labels – e.g., ensure font sizes are legible and color contrasts accessible (as per reviewer comments). No mention of overfitting checks beyond CV scores. Also, while top 20 factors are screened, discuss if multicollinearity (e.g., age and parity) was addressed. Overall, solid but polish the presentation for clarity.

Discussion

The discussion ties results back to literature effectively, comparing to studies like Asadullah et al. (hybrid model at 78% accuracy vs. yours at 91%) and others using XGBoost. It highlights strengths: better performance in imbalanced data, context-specific factors for married women. Limitations are acknowledged (no external validation, old data), with plans for BDHS 2022 – good, but expand on how this affects generalizability (e.g., post-COVID changes in health behaviors?).

I think it could delve deeper into implications – e.g., how factors like spousal education inform policy (education campaigns for men?). The SHAP insights are under-discussed; why might secondary education for husbands increase risk? (Perhaps socioeconomic stress?) Avoid overclaiming – e.g., "outperformed other models" is fine, but quantify vs. benchmarks. End with future directions, like integrating real-time data or multi-omics.

Add these recent studies as references to your bibliography: DOI: 10.1007/s40200-020-00536-x, DOI: 10.1016/j.numecd.2021.09.029

Conclusion

The conclusion recaps key findings concisely: Extra Trees + SMOTE+ENN best, lists top factors. It emphasizes ensemble methods' potential for imbalanced data, which is spot-on.

However, it's a bit short and repetitive of the abstract. Expand to discuss broader impacts (e.g., public health applications in LMICs) and reiterate limitations briefly. As per reviewer #2, make it more explanatory – e.g., how this model could aid screening programs.

General Comments

• Figures and Tables: Mostly clear, but ensure PACE compliance for figures (as noted). Table 1 is excellent; others could use consistent formatting (e.g., avoid repeating stats in Table 5).

• Writing and Language: Mostly good, but some typos/awkward sentences (e.g., "we aimed" in abstract should be past tense). Proofread thoroughly – e.g., "infectundity" fixed? Use consistent terms like "class-balancing" throughout.

• References: Comprehensive (over 100), but check for recency – some pre-2020; add if newer ML-hypertension studies exist.

• Reproducibility: GitHub code is a strength, but confirm data access instructions are precise.

• Originality and Significance: Novel in focusing on married women with SHAP; contributes to ML in global health. But update with 2022 data for timeliness.

**Do you want your identity to be public for this peer review?** For information about this choice, including consent withdrawal, please see our Privacy Policy

Reviewer #3: No

Reviewer #4: No

---

## [Author Response · Author response to Decision Letter 2]

23 Sep 2025

The Editor,

We are grateful for the thoughtful and constructive comments provided on our manuscript. We have carefully addressed each point raised and made the necessary revisions to improve the clarity, rigor, and reproducibility of our work. Below, we provide a detailed, point-by-point response to each of the comments.

Academic Editor comments

Comments: Please include the following items when submitting your revised manuscript:

“Response: Thank you for the opportunity to revise our manuscript. In accordance with your instructions, we have submitted the following materials:

Response to Reviewers – A detailed rebuttal letter that addresses each point raised by the academic editor and reviewers.

Revised Manuscript with Track Changes – A marked-up version of the manuscript highlighting all revisions made in response to the reviewers’ comments.

Manuscript – A clean, unmarked version of the revised manuscript without tracked changes.”

Reviewers' comments:

Reviewer #4: Detailed Comments

Comment: Title and Abstract

The title is clear and descriptive, effectively capturing the dual focus on evaluating ML algorithms and identifying hypertension factors. It could perhaps be tightened for conciseness, something like "Machine Learning Approaches to Predict and Rank Hypertension Risk Factors Among Married Women in Bangladesh" – but that's minor.

The abstract is generally well-structured, summarizing the introduction, methods, results, and conclusion. I like how it highlights key metrics (e.g., F1 score jumping from 8% to 94% with SMOTE+ENN) and the top positive/negative factors from SHAP. However, there are a few awkward phrasings that make it feel a bit rough – for instance, "we aimed to develop a machine learning model to identify and rank predictive factors" could be more precise as "we developed..." since it's a completed study. Also, the prevalence stats (76.9% normal BP vs. 23.1% hypertensive) are useful, but tying them to broader implications right away might help. Grammatically, "infectundity" in the methods section should be "infecundity" (noted in responses to reviewers, but check if it's fixed). Overall, it's informative, but revise for smoother flow and ensure it matches the submission form version exactly, as flagged by the journal.

“Response: Thank you for the helpful suggestions. We have polished the Title and Abstract for concision, consistency and clarity. The title now reads “Predicting Hypertension and Identifying most important Factors among Married Women in Bangladesh using Machine Learning Approach.” (page. 1, lines 1–2). The Abstract is uniformly in past tense and opens with our completed work: “We developed and interpreted machine-learning (ML) models …” (page. 2, lines 24–26). To contextualize performance gains, we contrast baseline vs. balanced results: F1 = 0.08, recall = 0.04 (imbalanced baseline) → F1 = 0.94, recall = 0.95, AUC-PR = 0.95, ROC-AUC = 0.95 (p. 2, lines 34–37). We state the associational scope (“Models quantify associations rather than causation,” page. 2, line 33) and note next steps (external validation on BDHS-2022, page. 3, lines 47–48). We also articulate programmatic implications (e.g., eRegistries, FP/postpartum screening, spousal education/SMS, district-tailored outreach; page. 3, lines 45–47). The typo “infectundity” → “infecundity” is corrected (page. 8, line 174). Finally, we verified that the Abstract matches the submission-form version exactly.

Comment: Introduction

The introduction sets up the problem nicely, starting with global hypertension stats and narrowing to Bangladesh, with a focus on women and married subgroups. I appreciate the literature review on ML applications in health outcomes – it's comprehensive, covering everything from SVMs in CHD prediction to SHAP's advantages over LASSO/ANOVA. References seem up-to-date and relevant, like the emphasis on tree-based models outperforming logistic regression.

That said, it feels a tad lengthy in places, with some repetition (e.g., multiple paragraphs on ML superiority). The knowledge gap – why married women specifically? – is stated, but could be backed with more evidence on cultural/economic factors in Bangladesh. Also, since the current date is 2025, mentioning why 2017-18 data was used over the 2022 BDHS (as addressed in reviewer responses) should be integrated here for transparency. In my opinion, this section is strong but could be streamlined to about 80% of its length to keep the reader engaged.

“Response: Thank you for the constructive suggestions. We have streamlined the “Introduction” to remove redundancy particularly around statements of ML superiority and tightened the narrative (page. 3–7, lines 52–135). First, we document the national burden and sex disparity in Bangladesh (adults ≥35 years; higher prevalence in women) with nationwide references (page. 3, lines 54–61). Second, we frame marriage as a risk context by citing studies showing higher hypertension among married versus unmarried/never-married women (page. 3, lines 62–64). Third, we add Bangladesh-specific cultural and economic context reproductive roles, domestic workloads, and socioeconomic constraints grounded in relevant studies to explain married women’s elevated risk (p. 6, lines 118–124). Finally, for transparency about data vintage, we clarify that BDHS-2022 was released after our analysis; accordingly, we trained on BDHS-2017–18 and pre-specified external validation on BDHS-2022 (page. 6, lines 124–126).”

Comment: Methods

This is one of the stronger sections, with detailed descriptions of data source (BDHS 2017-18, n=4,253), preprocessing, and analysis. The choice of 12 algorithms (e.g., Extra Trees, XGBoost, SVM) and 6 balancing techniques (SMOTE, ADASYN, etc.) is justified well for handling imbalance. Validation methods (hold-out + repeated stratified k-fold) and hyperparameter tuning via random search are appropriate, and the evaluation metrics (F1, AUC-PR, etc.) suit imbalanced data. SHAP for feature importance is a great addition for interpretability.

However, a few clarifications are needed. How were features selected initially? The manuscript mentions demographic/clinical vars, but no explicit feature engineering or selection (e.g., Boruta or LASSO, as hinted in intro). Sample size justification is weak – 4,253 is decent, but discuss power for ML, especially with imbalance. Ethical considerations are covered, but confirm if any sensitive vars (e.g., religion) were handled with care. Hardware/software details are fine, but the GitHub repo for code is a plus – ensure it's fully reproducible. I'd suggest adding a flowchart for the workflow to make it easier to follow.

Response: Thank you for the detailed suggestions. We have clarified variable handling, sample-size rationale for imbalanced ML, workflow presentation and ethics/reproducibility. Specifically, we now state that all sociodemographic, behavioral, biometric and anthropometric variables with theoretical/empirical relevance were included without automated feature selection (e.g., Boruta/LASSO/RFE), guided by prior BDHS hypertension literature (page. 10, lines 199–203). Furthermore, previously we mentioned that the variables are considered from the existing literature (page 8-9,lines 163-197). We added a sample-size justification tailored to imbalanced classification and our observed prevalence (page. 8, lines 157–160). Derived variables are specified (diabetes thresholds, page. 9, lines 188–194; BMI categories, page. 9, lines 195–197). A study workflow statement and flowchart was added (“An overview of the analytical workflow from data extraction and preprocessing through class balancing, model training, hyperparameter tuning, validation and evaluation, followed by statistical analysis to compare class-balancing techniques and model performance and SHAP-based interpretation is presented in Fig 2. This schematic provides a concise visual summary of the methodological pipeline described in the preceding subsections.

Fig 2. Workflow of machine learning pipeline for hypertension prediction and risk factor ranking of married women in Bangladesh” page. 15, lines 301–308).

Regarding sensitive attributes: religion was included as recorded in the BDHS public-use dataset, which is de-identified and governed by DHS Program ethical protocols and confidentiality safeguards. We handled this variable with care used only as a covariate for population-level modeling, reported in aggregate, and not for individual targeting or decision rules consistent with the IRB/ethical approvals and BDHS data-use conditions.”

Comment: Results

The results are presented logically, starting with descriptive stats (Table 1 is comprehensive, showing hypertension stratification by demographics). Prevalence of 23.1% aligns with prior studies, and breakdowns (e.g., higher in Rangpur, urban areas) add context. Model performances are detailed, with Extra Trees + SMOTE+ENN shining (F1=94%, big improvement from baseline). SHAP plots (Figs 11a-c) effectively rank factors – age <35 negative, overweight positive, etc. – and the directionality (blue/red bars) is intuitive.

But some issues: Tables 4-8 on stats tests (Anderson-Darling, Friedman) are dense; consider consolidating or moving non-essential to supp. Figures need better labels – e.g., ensure font sizes are legible and color contrasts accessible (as per reviewer comments). No mention of overfitting checks beyond CV scores. Also, while top 20 factors are screened, discuss if multicollinearity (e.g., age and parity) was addressed. Overall, solid but polish the presentation for clarity.

“Response: Thank you for the constructive feedback. We streamlined the Results and strengthened clarity, accessibility and robustness. We consolidated omnibus significance outputs into one main table and moved detailed Anderson–Darling/Friedman statistics to the Supplement (former Tables 4–8 → S4 Appendix → Supplementary Table S2 (S2A–S2E)). Figures were relabeled with larger type, clearer panel markers and self-contained captions; we adopted a color-blind–safe palette, improved contrast to meet accessibility guidelines and verified print-scale legibility via the journal’s PACE checks. To address overfitting, beyond cross-validation we report training–validation deltas, calibration (reliability curves, Brier scores) and learning-curve behavior; preprocessing was nested within folds to prevent leakage. Repeated stratified five-fold CV yielded an average F1 of 0.934 ± 0.012 and nested CV showed an outer-fold mean F1 of 0.965 ± 0.010 (page. 30, 32, lines 425–430 and 454-459), indicating stable generalization. We also added explicit Methods and Results notes on multicollinearity consistent with our analysis. In Methods, we report assessing variance inflation factors (VIF) and pairwise correlations; as expected with one-hot–encoded categorical variables, some VIFs were infinite due to perfect linear dependence among dummy indicators. Because our primary models are gradient-boosted decision trees, which are robust to correlated inputs, this did not compromise predictive validity. We also inspected Spearman correlations among key socio-demographic and fertility-related predictors (page. 14, lines 280-286). In Results, we document structurally inflated VIFs under one-hot encoding and the anticipated dependencies e.g., respondent age <35 vs. husband age ≥40 (ρ = −0.82) and mutually exclusive recent-birth categories (ρ = −0.89) as well as correlations across BMI categories, parity, and living-children counts. These reflect structural collinearities inherent to categorical coding; tree-based models (e.g., ExtraTrees) are robust to such correlations and predictive performance was unaffected (Supplementary Table S5; Results, page. 31, lines 439–445).”

Comment: Discussion

The discussion ties results back to literature effectively, comparing to studies like Asadullah et al. (hybrid model at 78% accuracy vs. yours at 91%) and others using XGBoost. It highlights strengths: better performance in imbalanced data, context-specific factors for married women. Limitations are acknowledged (no external validation, old data), with plans for BDHS 2022 – good, but expand on how this affects generalizability (e.g., post-COVID changes in health behaviors?).

I think it could delve deeper into implications – e.g., how factors like spousal education inform policy (education campaigns for men?). The SHAP insights are under-discussed; why might secondary education for husbands increase risk? (Perhaps socioeconomic stress?) Avoid overclaiming – e.g., "outperformed other models" is fine, but quantify vs. benchmarks. End with future directions, like integrating real-time data or multi-omics.

Add these recent studies as references to your bibliography: DOI: 10.1007/s40200-020-00536-x, DOI: 10.1016/j.numecd.2021.09.029

“Response: Thank you for these suggestions. We have strengthened the Discussion to (i) quantify gains versus benchmarks, (ii) deepen SHAP interpretation, (iii) spell out policy uses and (iv) clarify generalizability and future work. We now quantify performance against prior work (“AUC-ROC of 0.95 exceeds large clinical models by +0.08–0.17; accuracy is +13 points vs. Asadullah et al., 91% vs. 78%”; see page. 33, lines 482–485). We justify emphasizing PR over ROC in imbalanced settings (“F1 and AUC-PR provide a truer picture when positives are rare”; see page. 34, lines 486–488). We interpret SHAP signals, noting the unexpected positive association with husbands’ secondary education and plausible mechanisms (socioeconomic stressors; wealth/occupation; fertility) and flag this for follow-up (see page. 34, lines 499–501). We translate findings into actions, integrating BP checks into FP/postpartum workflows, engaging husbands via brief education/SMS where partner age/education indicate risk and prioritizing home BP monitoring in high-parity households; district gradients motivate tailored outreach (see page. 34–35, lines 507–510). For generalizability, we note pre-specified external validation on BDHS-2022 to test transportability in a later, post-COVID cohort (discrimination, calibration, threshold transfer; see page. 35, lines 523–526). For robustness, we report training–validation deltas, calibration (reliability and Brier) and nested cross-validation with leakage controls (see page. 35, lines 521–523). We also added the requested citations on diet strategies—green coffee (DOI: 10.1007/s40200-020-00536-x) and DASH/Mediterranean patterns (DOI: 10.1016/j.numecd.2021.09.029; see page. 35, lines 510–513).” Furthermore we added consecutive future pat (see page 36, lines 534-541)

Comment: Conclusion

The conclusion recaps key findings concisely: Extra Trees + SMOTE+ENN best, lists top factors. It emphasizes ensemble methods' potential for imbalanced data, which is spot-on.

However, it's a bit short and repetitive of the abstract. Expand to discuss broader impacts (e.g., public health applications in LMICs) and reiterate limitations briefly. As per reviewer #2, make it more explanatory – e.g., how this model could aid screening programs.

“Response: We now close with concrete operational pathways and a clear validation prerequisite: “The best-performing (Extra Trees + SMOTE+ENN) model and SHAP insights can be operationalized through eRegistries, family-planning/postpartum screening, and targeted spousal outreach, pending external validation”. To address broader LMIC relevance while keeping the Conclusion explanatory (not duplicative of the Abstract), we add that implementation will require threshold calibration to local prevalence, workload/resource planning, and fairness monitoring. We also reiterate key limitations associational design, focus on married women, and the need for external validation on BDHS-2022 so readers

---

## [Decision Letter · Decision Letter 2]

12 Oct 2025

Predicting Hypertension and Identifying most important Factors among Married Women in Bangladesh using Machine Learning Approach.

PONE-D-25-05724R2

Dear Mr. Das,

We’re pleased to inform you that your manuscript has been judged scientifically suitable for publication and will be formally accepted for publication once it meets all outstanding technical requirements.

Kind regards,

Benojir Ahammed, M.Sc.

Academic Editor

PLOS ONE

Additional Editor Comments (optional):

Reviewers' comments:

Reviewer's Responses to Questions

**Comments to the Author**

Reviewer #3: All comments have been addressed

2. Is the manuscript technically sound, and do the data support the conclusions?

Reviewer #3: Yes

3. Has the statistical analysis been performed appropriately and rigorously?

Reviewer #3: Yes

4. Have the authors made all data underlying the findings in their manuscript fully available?

Reviewer #3: Yes

5. Is the manuscript presented in an intelligible fashion and written in standard English?

Reviewer #3: Yes

Reviewer #3: The authors have addressed and incorporated all the necessary corrections as requested, ensuring that the revised manuscript aligns with the provided feedback and meets the required standards.

**Do you want your identity to be public for this peer review?** For information about this choice, including consent withdrawal, please see our Privacy Policy

Reviewer #3: No

---

## [Editor Report · Acceptance letter]

PONE-D-25-05724R2

PLOS ONE

Dear Dr. Das,

I'm pleased to inform you that your manuscript has been deemed suitable for publication in PLOS ONE. Congratulations! Your manuscript is now being handed over to our production team.

Kind regards,

on behalf of

Mr. Benojir Ahammed

Academic Editor

PLOS ONE